# Threshold concentration and random collision determine the growth of the huntingtin inclusion from a stable core

Sen Pei [1], Theresa C. Swayne[2], Jeffrey F. Morris [3,4] & Lesley Emtage [1,5✉]

The processes underlying formation and growth of unfolded protein inclusions are relevant to neurodegenerative diseases but poorly characterized in living cells. In *S. cerevisiae*, inclusions formed by mutant huntingtin (mHtt) have some characteristics of biomolecular condensates but the physical nature and growth mechanisms of inclusion bodies remain unclear. We have probed the relationship between concentration and inclusion growth in vivo and find that growth of mHtt inclusions in living cells is triggered at a cytoplasmic threshold concentration, while reduction in cytoplasmic mHtt causes inclusions to shrink. The growth rate is consistent with incorporation of new material through collision and coalescence. A small remnant of the inclusion is relatively long-lasting, suggesting that it contains a core that is structurally distinct, and which may serve to nucleate it. These observations support a model in which aggregative particles are incorporated by random collision into a phase-separated condensate composed of a particle-rich mixture.

[1] Biology Department, City University of New York, York College, Queens, NY, USA. [2] Herbert Irving Comprehensive Cancer Center, Columbia University, New York, NY, USA. [3] Department of Chemical Engineering, The City College of New York, New York, NY, USA. [4] Benjamin Levich Institute, The City College of New York, New York, NY, USA. [5] Molecular, Cellular and Developmental Biology Program, City University of New York Graduate Center, New York, NY, USA. ✉email: lemtage@york.cuny.edu

Deposits of unfolded protein are characteristic of neurodegenerative disorders, including Huntington's disease. Visible deposits in the brains of patients with Huntington's disease are predominantly composed of an N-terminal cleavage product of mutant huntingtin (mHtt) protein[1,2]. Causative mutations expand a polyglutamine (polyQ) repeat tract near the N-terminus; in mice, expression of N-terminal fragments of mHtt leads to accumulation in cytoplasmic and nuclear inclusions, recapitulating many aspects of Huntington's disease[3–5].

While mHtt is expressed in most tissues throughout life, cell death due to mHtt is restricted to particular regions of the brain. Most cells, therefore, are able to cope with persistent expression of moderate, or even high levels of mHtt protein, despite its intrinsic instability. To shed light on the underlying cause of death in cells that succumb, it would be useful to have a better understanding of the mechanisms by which most cells are able to successfully process sustained, high levels of unfolded protein. We study the response of cells to a constant burden of unstable protein using GFP-fused Htt from the constitutive GPD promoter in *S. cerevisiae*.

When expressed in cultured cells, exon 1 of mHtt, fused to a fluorescent protein or visualized with an antibody, forms cytoplasmic inclusions[6,7]. Expressed in yeast, mutant Httex1-GFP typically forms a singular, ovoid inclusion body, and small, moving aggregative particles[8–10]. Mutant Httex1 inclusions in yeast have been characterized as insoluble protein deposits (IPOD) whose contents do not exchange with the surrounding cytosol and which receive new material through active transport[11–14]. However, the IPOD model of the mHtt inclusion is not consistent with the more recent finding that mutant Httex1-GFP in *S. cerevisiae* can diffuse throughout the inclusion body, or that mHttex1-GFP is released from the inclusion body, indicating that the contents exchange with the cytoplasm. The fluorescence recovery time of photobleached inclusion bodies is markedly slower than those seen in liquid-liquid phase-separated compartments, but are consistent with diffusion times in gels[8].

In addition to the experimental evidence indicating that material inside the inclusion body is not fixed in place, time-lapse imaging demonstrated that inclusions themselves are mobile. While the motion of the inclusion bodies is mildly superdiffusive, it has no apparent direction, and the smaller aggregative particles of mHttex1 exhibit purely Brownian motion[8]. Deletion of either the low-complexity protein *RNQ1* or the aggregase/disaggregase *HSP104* prevents the formation of both small particles and inclusions. The presence of a large, mobile phase-separated inclusion, accompanied by smaller particles of unfolded protein diffusing through the cytoplasm, suggests that the inclusion forms through the coalescence of particles. However, supportive evidence for this model has been lacking.

In solution, liquid-gel or liquid-liquid phase separations show a sharp concentration dependence[15–17]. The model of the mHttex1 inclusion body as a phase-separated condensate that grows by collision and coalescence with small particles of unfolded protein predicts that inclusion body formation should show a concentration dependence. At very low cytoplasmic concentrations, inclusion bodies should diminish. Additionally, if the material is accumulated through collision with the inclusion surface, we would predict that the growth rate of the inclusion body volume should increase with the surface area, and it should also be modulated by cytoplasmic concentration.

We examined the concentration dependence of inclusion body formation in vivo and found that there appears to be a threshold concentration below which inclusion bodies do not form. Conversely, when we used auxin-mediated degradation to reduce cytoplasmic mHttex1-GFP to low levels, material was lost from inclusion bodies. We observed that there appears to be a small kernel of mHttex1-GFP that persists more stably, with a rate of loss that is markedly lower. Surprisingly, we also found that the proteasome is not typically overwhelmed in cells that form inclusions, indicating that it has a similar capacity to degrade mHttex1-GFP in cells with and without inclusion bodies.

The inclusion growth rate is consistent with the model in which material is absorbed after the collision of aggregative particles with the surface, and is inconsistent with models suggesting that new material is sent to inclusion bodies primarily through active transport (Fig. 1)[11,12,14,18]. We conclude that rising concentrations of unfolded protein trigger the nucleation of an inclusion body by the cell. Growth of the inclusion occurs through collision and coalescence of small, aggregative particles into a suspension of particles that is phase-separated from the surrounding cytoplasm. These studies constitute a detailed analysis of the concentration dependence and growth of a phase-separated body in vivo, and suggest that the process is nucleated and occurs prior to a significant reduction in the capacity of the ubiquitin-proteasome system (UPS).

## Results

### Time required to initiate an inclusion body is not strongly dependent on mHtt concentration.

We measured the time required to initiate a mHttex1-GFP inclusion body and inclusion growth. Inclusion bodies composed of mHttex1-GFP are typically singular and do not appear to fission or fuse. In order to confirm that we would be able to track individual inclusions over long time periods, we selectively imaged 11 atypical cells with two (in one case, three) mHttex1 (72Q)-GFP inclusion bodies over a cumulative total of 35.7 h, imaging every 10 min. We did not observe the fusion of inclusion bodies. Nor have we ever observed fission of inclusion bodies when imaging at 10-min intervals, or when imaging for shorter durations at intervals of ~30 ms. Thus, we are able to follow individual inclusion bodies over several cell cycles.

In order to determine the relationship between the time required for a mHttex1-GFP inclusion body to form and the cytoplasmic mHtt concentration, we used cells expressing

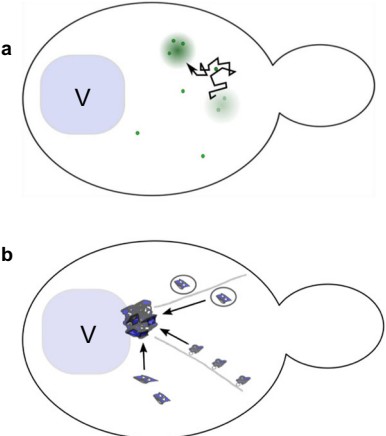

**Fig. 1 Two proposed models for the growth of the mutant Htt inclusion in *S. cerevisiae*. a** Model of mHtt inclusion as mobile phase-separated body: small particles move randomly in the cytoplasm, and are integrated into the inclusion through collision and coalescence. **b** Model of mHtt inclusion as an IPOD: the IPOD is described as docked at the vacuolar membrane. Active transport along actin cables has been shown to be required for normal IPOD growth, and it has been proposed that material is transported in Cvt vesicles or mediated by Myo2. Transport by currently unknown mechanisms is also speculated to contribute to IPOD growth. V vacuole.

mHttex1(72Q)-GFP from a low copy number (CEN) plasmid, p415, under the control of the GPD promoter. The p415 plasmid has been found to be present at typically two–five copies per cell in our parental strain, BY4741[19]. Variations in plasmid copy number result in variable cytoplasmic concentration of mHttex1 (72Q)-GFP protein. In order to ensure uniformity of culture and imaging conditions, we used cells from a single mid-log phase culture, grown under identical conditions but with variable cytoplasmic concentrations of mHttex1-GFP, to measure inclusion body formation and growth.

As mHttex1-GFP is continuously synthesized over the imaging timecourse, an individual cell contains a collection of GFP molecules that have been exposed to a variable number of exposures. As a result, it is not possible to correct mHttex1-GFP fluorescence intensities accurately for photobleaching over long timecourses, due to continuous synthesis of mHttex1-GFP[8]. To minimize any effect of photobleaching in this experiment, we imaged each field of cells only once per hour.

Forty individual cells growing on agar pads and expressing mHttex1-GFP were imaged every hour over a period of 5–6 h. The original cells continued to grow and divide during the entire imaging period (Supplementary Fig. 1), with a doubling time of $130 \pm 7$ min (mean $\pm$ SEM, $n = 27$ initial cells), somewhat longer than the doubling time for the same cells in liquid culture with shaking ($106 \pm 2$ min, $n = 10$ cultures).

We tracked both cytoplasmic levels of mHttex1-GFP and the formation of inclusion bodies in a total of 152 cells: the 42 initial cells and 110 of their progeny. Of those, 13 cells had an inclusion at the beginning of the experiment, and 80 formed inclusion bodies during the timecourse, while 59 cells did not form an inclusion during the period of our observations. Figure 2 shows a timecourse of a typical cell dividing over 6 h to give rise to six additional daughter cells.

The cytoplasmic intensities of individual cells remained fairly constant over the course of the experiment (Supplementary Fig. 2). In order to establish whether there is a relationship between the cytoplasmic concentration of mHttex1-GFP and the time to inclusion body formation, we began by measuring the time from the birth of a cell to the formation of an inclusion body. Our data set was restricted to cells that formed an inclusion body and were born during the course of the experiment (Fig. 2b). Cells present at the beginning of the experiment were excluded because we could not know how long those cells existed prior to the onset of imaging. For cells that were born during the course of the experiment and formed an inclusion body, the average time to inclusion formation was $1.6 \pm 0.1$ h (mean $\pm$ SEM, $n = 56$) (Fig. 2c, d).

Although there appears to be a weak inverse relationship between cytoplasmic concentration and the average time to form an inclusion body in those cells that do form one, the correlation was not significant ($n = 56$ cells, $p = 0.11$). If one considers the cells with cytoplasmic mHttex1-GFP intensity in the lowest quintile, the average time to inclusion body formation was $1.5 \pm 0.4$ h ($n = 11$), while the average time was $1.1 \pm 0.1$ h ($n = 11$) for cells with cytoplasmic intensity in the highest quintile (Fig. 2d).

**Cells require a threshold cytoplasmic mHtt concentration to form an inclusion body**. Is there a concentration threshold required for inclusions to form? We monitored the intensity of cytoplasmic mHttex1-GFP over the timecourse. While inclusion bodies formed in a wide range of cytoplasmic mHtt concentrations, cells that failed to form an inclusion often had very low cytoplasmic mHtt concentrations. Of inclusion-forming cells, 81% formed inclusions within 2 h of the birth of the cell. Nine of the eleven cells in the lowest quintile of cytoplasmic concentration formed an inclusion in ≤2 h. In our analysis of cells which

did not form an inclusion body, we considered only cells that were born over 2 h before the end of the experiment, which were observed for 2 or more hours after the birth of the cell (Fig. 3a, b). Using this criterion, we identified 37 cells that did not form an inclusion during the course of the experiment, including five of the original cells present at the beginning of the imaging session, and 32 cells born during the experiment. These noninclusion-forming cells were observed from 2–6 h, with a mean observation time of $3.6 \pm 0.2$ h ($n = 37$, SEM).

Our data are consistent with a concentration threshold of mHttex1-GFP (Fig. 3c) for inclusion body formation in growing cells. No cell with an average cytoplasmic concentration under 600 AU formed an inclusion during the period of our observations. Conversely, all but 2 of 102 cells with a cytoplasmic intensity >1000 formed inclusions. The average cytoplasmic intensity in inclusion body-forming cells was $1485 \pm 90$ AU ($n = 55$, SEM), ranging from 605–3194.

Httex1 stability is reduced at higher polyQ tract lengths, which should result in a shift of the threshold for inclusion body formation to a lower value. We used cells expressing mHttex1 (103Q)-GFP to measure the threshold for inclusion body formation, and found a large overall reduction in the cytoplasmic intensity at which inclusion bodies will form (Supplementary Note 1 and Supplementary Figs. 3, 4).

**At low cytoplasmic mHttex1-GFP concentrations, inclusion bodies shrink**. The threshold cytoplasmic mHttex1-GFP intensity for inclusion body formation suggests that there is a concentration threshold required for either the initiation or the growth of an inclusion. Previous studies demonstrated that the inclusion body is a phase-separated, although not liquid, compartment[8]. Phase separation is concentration-dependent and reversible in vitro. Thus, we next asked what would happen if the cytoplasmic concentration of mHttex1-GFP were to drop in vivo. We used auxin-inducible degradation (AID) to drive down cytoplasmic mHtt levels and observe the effect on inclusion body size. AID was conferred by fusing the degron sequence (IAA$^{71–114}$) to mHttex1-GFP together with co-expression of the plant E3 ubiquitin ligase Tir1. Tir1 binds to the degron sequence in the presence of the plant hormone auxin (1-naphthalene acetic acid (NAA)), resulting in the ubiquitination of the degron and degradation of the target protein[20,21]. Incorporation of the degron sequence did not significantly affect inclusion formation by Httex1, and the presence of the E3 ligase Tir1 did not affect the frequency of degron-tagged mHtt inclusions in the absence of auxin (Supplementary Fig. 5).

Cells expressing Tir1 and mHttex1(72Q)-degron-GFP were imaged every 15 min for 4 h by spinning-disk confocal microscopy. The addition of auxin caused a substantial drop in cytoplasmic intensities of cells expressing mHttex1-degron-GFP (Fig, 4a). As a control, we treated cells with the identical volume of vehicle (95% ethanol), which had no effect on cytoplasmic intensity (Supplementary Fig. 6).

To study the fate of inclusion bodies during inducible degradation of mHtt, we identified cells containing inclusion bodies of moderate to large size (maximum diameters 0.6–1.0 μm) that also showed a decrease in cytoplasmic intensity to the background by about the second hour of imaging. The volume of the inclusion body was measured over the 4-h timecourse. Empirical correction of the cytoplasmic intensity for photobleaching will substantially overcorrect as new material is continuously synthesized during the timecourse[8]. However, once the cytoplasmic concentration of mHttex1-degron-GFP is very low, new material can no longer be added to the inclusion body in appreciable quantities. Thereafter, the collection of molecules in

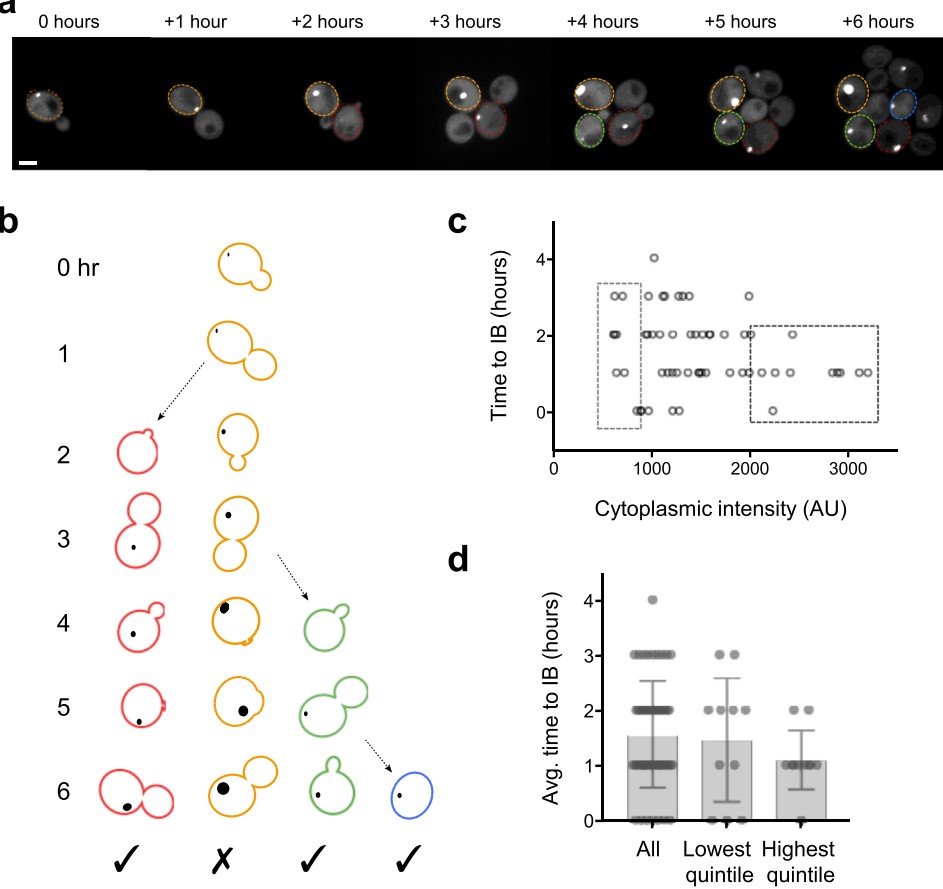

**Fig. 2 Time to the formation of inclusion bodies is not strongly correlated with cytoplasmic intensity. a** A typical field of cells imaged over a 6-h timecourse is shown; the maximum projection includes sections containing inclusion bodies. Outlines indicate the cells shown in **b**. The contrast has been set to identical parameters for all images. Scale bar, 2 μm. **b** Diagram illustrating criteria for inclusion in this analysis, showing the original cell in **a** and three cells born during the experiment. For the analysis in **c** and **d**, measurements were taken only from cells that were born during the experiment and formed a new inclusion body (checkmark). **c** Scatter plot of cytoplasmic GFP intensity vs time elapsed between the birth of a cell and formation of an inclusion body (IB). There is a slight non-significant correlation between early birth and high intensity ($n = 56$, Pearson's $r = -0.22$, $p = 0.11$). Boxes indicate cytoplasmic intensities in the lowest and highest quintiles. **d** Time to IB formation for cells in the lowest and highest quintiles of cytoplasmic intensity, and for all cells analyzed (mean ± SD, $n = 56, 11, 11$). The difference between times to IB formation is not significant (Kruskal–Wallis test across all three categories, $p = 0.32$; Mann–Whitney test between all IBs and the highest quintile, two-tailed distribution, $p = 0.12$).

the inclusion body can be accurately corrected for photobleaching. Therefore, we have corrected all intensity data for photobleaching, but note that the corrected intensities prior to the loss of cytoplasmic fluorescence may be slightly inflated. However, we are most interested in the period after cytoplasmic levels dropped close to the background, at which point correction for photobleaching is more accurate.

Inclusion body sizes vary widely from cell to cell at the beginning of the experiment. In order to compare the amount of material in inclusion bodies, we normalized the volume and integrated density (ID) of each inclusion to its maximum and aligned the measurements at the point of maximum volume (Fig. 4b), or ID, which is the sum of the intensity values within a region of interest, reflecting the quantity of protein in the inclusion. (Supplementary Fig. 7). Cytoplasmic intensities were also aligned to the point of maximum inclusion body intensity, in order to determine the approximate relative cytoplasmic levels at which inclusion body growth reverses.

Under the imaging conditions used in this experiment, cytoplasmic levels of mHttex1-degron-GFP were 5–65 AU above background prior to the addition of auxin; the mean intensity of the cytoplasm was 22.5 ± 2.5 ($n = 25$, SEM). Generally, inclusion bodies continued to grow as long as the cytoplasmic intensity of

mHttex1-degron-GFP remained at or above ~4–6 AU, regardless of the presence of auxin. For all inclusions, at low cytoplasmic intensities, the volume rose quickly to a peak, then underwent a sharp decline when the cytoplasmic intensity of mHttex1-degron-GFP fell close to background levels. Similar results were observed when ID was plotted over time (Supplementary Fig. 7).

Not all types of Httex1 inclusions respond to changes in cytoplasmic mHttex1 levels. To probe the structure and relationship between cytoplasmic mHttex1 and inclusions that lack the characteristics of phase separation, we used a mHttex1-degron construct lacking the proline-rich domain, which is important for mHttex1 stability in vitro and in vivo[22–27]. Inclusions formed from mHttex1ΔPRD-degron-GFP were insensitive to cytoplasmic mHttex1 levels (Supplementary Note 2, Supplementary Fig. 8, and Supplementary Movies 1, 2).

**The inclusion body has a stable core which serves to nucleate growth.** Moderate and large inclusion bodies shrank relatively quickly upon the addition of auxin, dropping to only 30% of their peak volume in 2 h. In contrast, we observed that inclusion bodies that were already small at the beginning of the experiment initially lost material rapidly, but that a small, fluorescent core persisted until the end of the experiment, up to several hours.

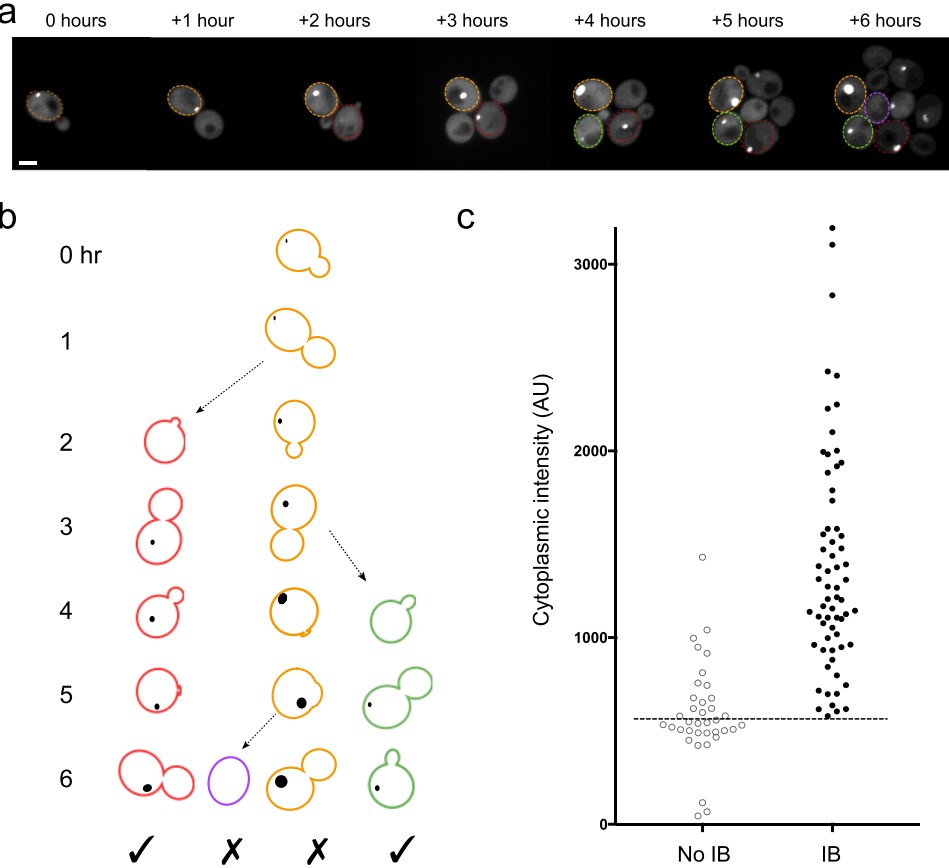

**Fig. 3 A threshold cytoplasmic concentration is required for inclusion body formation. a** The timecourse is shown in Fig. 2, but here the outlined cells indicate the four cells shown in **b**. **b** In this analysis, we considered only cells that acquired an inclusion body (IB) during the experiment and cells that did not acquire an IB but were born at least 3 h before the end of the experiment. **c** Each data point reports the average cytoplasmic intensity of a cell that formed an IB (IB, $n = 65$) or did not form an IB (no IB, $n = 37$) during the course of the experiment. The intensities of the two groups are significantly different ($p < 0.0001$, unpaired $t$-test, two-tailed distribution assuming unequal variance).

This core particle appeared to shrink at a much slower rate than the bulk of the inclusion (Fig. 5a).

We tracked the core particles of 11 small inclusion bodies whose cytoplasm reached background levels ≤2 hr after the addition of auxin. In order to compare the loss of material from persistent core particles with the loss of material from moderate to large inclusion bodies, we plotted ID, as the apparent size of the inclusion body core particles was well below our resolution. Of the 11 small inclusion bodies, none completely disappeared by the end of the timecourse, regardless of initial size (Fig. 5b). The initial loss rate was similar to that in moderate and large inclusion bodies (Fig. 5c). In contrast, at later stages, the average ID of the kernel was relatively steady (Fig. 5d).

To ascertain if the core particle would serve as a nucleus for regrowth of inclusions, we imaged 72Q-degron-expressing cells every 30 min for 5 h after washing out auxin. The cytoplasmic intensity remained low for ~1 h after washout. We tracked the response of ten inclusion bodies and 16 core particles to auxin washout (Fig. 6). The preexisting inclusion bodies began to grow again as cytoplasmic GFP levels increased; no cells were observed with two inclusion bodies. The cytoplasmic intensity threshold for regrowth was identical to the threshold at which they began to shrink, about 6 AU (Fig. 6b).

Of the 16 core particles, nine remained approximately the same size for 2.5–4 h before resuming growth (Fig. 6b), whereas five disappeared after 1–2.5 h. Two, which were in cells that regained very little cytoplasmic mHttex1-GFP, remained small but visible until the end of the timecourse, unchanged in apparent size for at

least 6 h. Every core particle served as a nucleus for the addition of new material; no cell formed additional inclusion bodies.

**UPS has excess capacity in cells that form inclusions**. Because inhibition of the proteasome can lead to the formation of inclusions, it has been proposed that inclusions form in response to an overload of the proteasome[28–30]; that is, the failure of one pathway of removal may lead to the formation of large aggregates of unfolded protein. Alternatively, blockage of the proteasome could allow cytoplasmic levels of unfolded protein to rise over a threshold required to trigger the formation of an inclusion body. We were interested in ascertaining whether the proteasome was overwhelmed by unfolded protein in cells that had formed inclusion bodies or whether it still had excess capacity.

Consistent with our previous data, the cytoplasmic mHttex1-GFP intensity appeared, on average, higher in cells with inclusion bodies (Supplementary Fig. 9). In a random selection of cells with and without inclusions, and subjected to auxin-induced mHtt degradation, we observe that cytoplasmic mHttex1-degron-GFP levels decrease substantially over the course of the experiment in almost all cells: cytoplasmic mHttex1-GFP levels fell in 100% of cells without inclusion bodies ($n = 86$), and in 98% of the cells with ($n = 43$). In cells with inclusion, the cytoplasmic signal remained visible for a longer period, but it dropped over the course of the experiment. However, we reasoned that the longer duration of detectability may have been the result of higher initial levels.

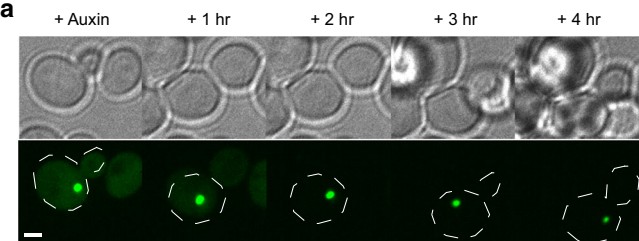

**a**

+ Auxin    + 1 hr    + 2 hr    + 3 hr    + 4 hr

**b**

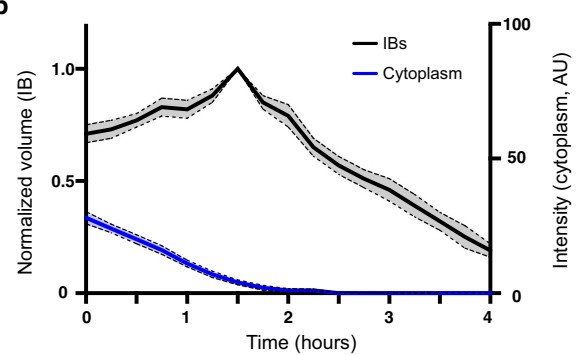

**Fig. 4 Mutant Htt inclusion bodies shrink when cytoplasmic mHtt levels drop to low levels. a** Time-lapse images of cells expressing Tir1 and mHttex1-degron-GFP were acquired immediately after the addition of NAA, and every 15 min for 4 h thereafter. Brightfield images (upper panels) and maximum-intensity projections of the GFP channel (lower panels) were taken shortly after the addition of auxin, then 1, 2, 3, and 4 h later, illustrating the loss of cytoplasmic GFP followed by decline in inclusion body (IB) size. The displayed contrast of the fluorescence images is identical across the timecourse. The cell boundary is indicated by a dotted white line. Scale bar, 2 μm. **b** Quantitation of average IB volume (black) and cytoplasmic intensity (blue) over time. Volume traces were normalized to the maximum for each IB, and aligned in time so that the maximum intensity of each individual IB occurs at the same time point ($n = 7-19$ cells per timepoint; shaded regions indicate SEM).

In order to compare the rate of auxin-induced cytoplasmic mHttex1-GFP loss in cells with and without inclusion bodies, we selected cells with initial cytoplasmic levels between 5–45 AU that contained inclusion bodies and compared them to cells with similar initial cytoplasmic levels, but no inclusion bodies, from the same imaging fields. Of 12 cells with moderate to large inclusion bodies, one did not lose cytoplasmic intensity; the other 11 lost cytoplasmic mHttex1-degron-GFP (Fig. 7a, b). The rate of mHttex1-GFP loss from the cytoplasm in both cells with and without inclusion bodies was very similar: although the decrease in cytoplasmic intensity in cells with an inclusion body was slightly slower, the difference was not significant (Fig. 7c). Since AID is mediated by the ubiquitin-proteasome system (UPS), these results are consistent with the supposition that the presence of an inclusion body is not associated with lower UPS capacity.

**Inclusion body growth rate increases with the surface area**. Two fundamentally different models of inclusion growth in yeast have been proposed. Mutant Htt is most commonly described as accumulating in an IPOD, which has been proposed to receive a substantial fraction of new material through active transport (Fig. 1b)[11,12,14,18]. However, the mHtt inclusion body is mobile, making it difficult to transport material to it via existing cytoskeletal networks. Also, the movement of visible small particles is not directional. Therefore, we have proposed that the mHtt inclusion body incorporates material through random collisions and coalescence.

Regardless of the mechanism, we expect the growth rate of the inclusion to be proportional to the concentration of unfolded, aggregated mHtt, which, in turn, we expect should be proportional to the cytoplasmic concentration of mHtt. However, the mechanism of growth is predicted to affect inclusion body growth kinetics. A model (Fig. 1a) in which growth occurs through the incorporation of small particles onto the surface predicts that the rate of growth will increase with surface area, but may be modulated by the formation of a concentration gradient of diffusing particles. In contrast, the growth of IPODs has been shown to require functioning active transport systems; if the material is actively transported to the inclusion, it suggests that growth may be dependent on the carrying capacity of the transport systems and independent of inclusion size. In the most extreme case, if the transport is limiting, the transport model predicts a constant rate of growth over time as long as the cytoplasmic concentration of aggregates remains constant.

If aggregative particles are incorporated into the inclusion body through collision, followed by incorporation, the growth rate depends on the surface area of the inclusion body and the speed with which particles are incorporated into it. The volume grows faster as the surface area increases because the frequency of productive collisions with small aggregative species increases with surface area. As cytoplasmic mHttex1-GFP has been observed to be continuously replenished by new protein synthesis[8], and to remain relatively constant with time (Supplementary Fig. 2), it would be reasonable to predict that the concentration of mHtt particles will also be relatively constant over time.

Using those assumptions, there are two possible outcomes: if a stable concentration gradient of mHtt particles develops around the inclusion body, we would expect growth to be limited by diffusion. In this case, we would predict that the surface area will grow linearly with time:

$$r^2 = r_0^2 + 2kc_\infty t \qquad (1)$$

where $c_\infty$ is the bulk cytoplasmic concentration of mHtt particles, and $k$ is the reaction constant, and $r$ is the radius of the inclusion body (see Methods).

However, if the rate of incorporation into the inclusion is slow and diffusion is sufficiently rapid that a concentration gradient of particles around the inclusion body does not form, the concentration at the inclusion body surface is the same as the bulk cytoplasmic concentration, and then we would predict that for any particular concentration of mHtt particles, the rate of growth of the radius will be constant over time:

$$r = r_0 + kc_\infty t \ \text{ or } \ \frac{dr}{dt} = kc_\infty \qquad (2)$$

In the case of active transport, if the amount of material received by the inclusion body is dependent on, and limited by, the volume of material transported to the inclusion body, then the rate of change in volume over time can be expressed as:

$$\frac{dV}{dt} = k_{tr}c_\infty \qquad (3)$$

Here the identity of the aggregative species has not been defined and is not necessarily related to the visible diffusing mHtt particles. The rate constant is defined by the carrying capacity of the transport mechanisms.

In order to evaluate models of inclusion formation and growth, we measured the growth of inclusions over time, using the same long-term imaging data set that we used to relate cytoplasmic intensity to the formation of inclusions. We restricted our data set to inclusions that were >0.3 μm in diameter and were followed for 4 or more hours ($n = 25$). The initial diameter of these inclusion bodies was 0.48 ± 0.14 μm (mean ± SD). For each inclusion body,

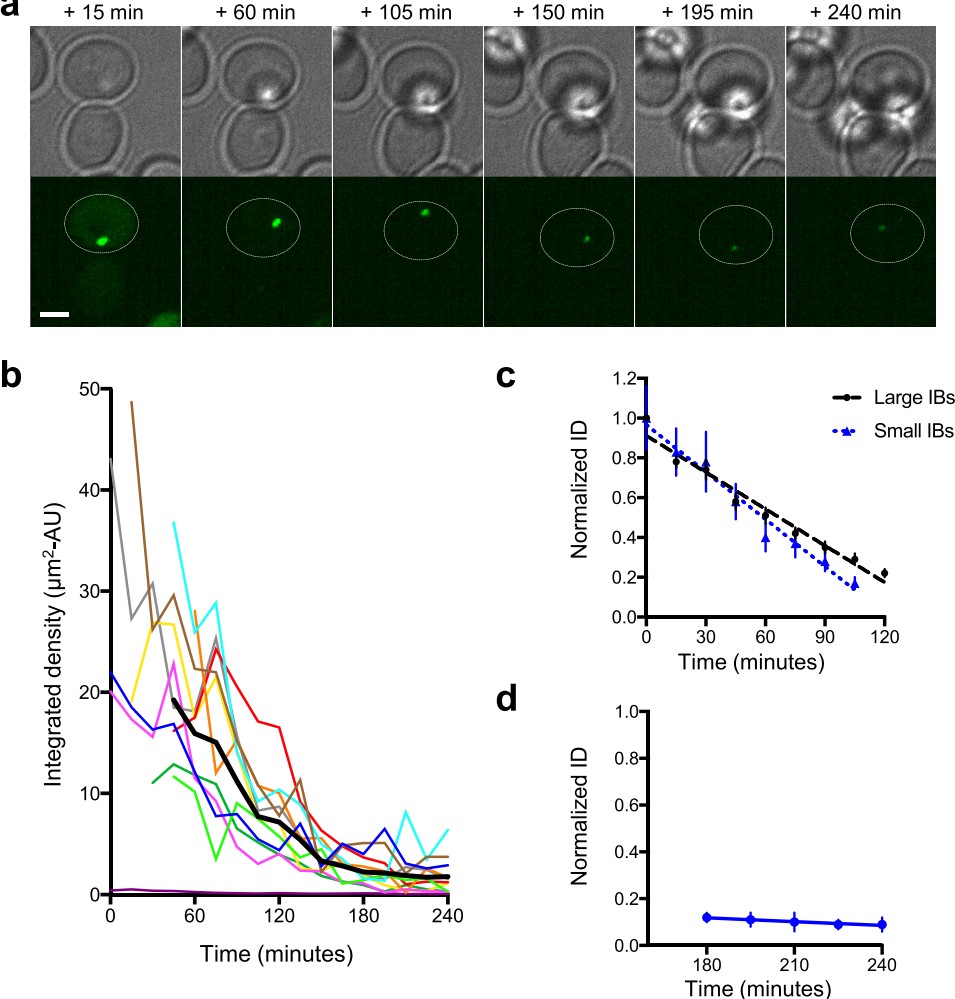

**Fig. 5 Biphasic loss of mHtt-GFP from the inclusion body reveals a persistent core. a** Brightfield (upper panels) and maximum-intensity projections of the GFP channel (lower panels) showing a cell containing a small mHttex1(72Q)-degron-GFP inclusion after the addition of auxin. Time after addition of auxin is shown. The cell containing the inclusion body (IB) is outlined. Scale bar, 2 μm. **b** Integrated density was plotted against time for 11 individual small IBs. The thick black line shows the average integrated intensity over time. IB intensities were corrected for bleaching. **c** The change in integrated density (ID) of small inclusions shown in **b** is shown for 19 moderate and large (black) and 11 small (blue) inclusions. The rate of decrease in ID is 37% hr$^{-1}$ for large IBs ($R^2 = 0.97$), and 47% hr$^{-1}$ for small IBs ($R^2 = 0.97$). Values are corrected for bleaching and normalized to initial IB intensity. **d** The rate of decrease in the ID of small IBs during the last 75 min of the timecourse is shown (3% hr$^{-1}$, $R^2 = 0.94$).

we estimated the volume from the cross-sectional area at the plane of maximum intensity at each time point.

A graph of all 25 inclusion bodies tracked for 4–6 h shows that the growth rate increased with increasing average cytoplasmic levels of mHtt (Fig. 8a). For example, comparing several inclusions with similarly large size at the beginning of imaging (≈0.18–0.30 μm$^3$), it is clear that inclusions in cells with high cytoplasmic intensity (1600–1700) grew faster than those in cells with low cytoplasmic intensity (860–1270). These data confirmed our expectation that the amount of unfolded mHttex1-gfp taken up by the inclusion body is proportional to the overall cytoplasmic concentration of mHttex1-GFP.

To reduce noise in our analysis of growth rate, we averaged the volume, cross-sectional area, or radius of all inclusion bodies. We fitted growth of volume, area, and radius with time as a straight line and found that the growth of the inclusion body radius was the best fit for a straight line with time, with the growth of the area with time also well-fit, either in the subset of cells with moderate or high cytoplasmic intensities ($n = 16$), or all cells (Fig. 8b and Supplementary Fig. 10). If we extend our analysis to all possible implied relationships between the geometries with

time, linear growth of the radius with time produces the best overall fits (Supplementary Table 1). These results are consistent with a model in which growth is limited neither by the amount of unfolded protein nor by the rate of transport, but rather growth accelerates proportional to the surface area of the inclusion itself.

Additionally, the linear relationship between growth and inclusion body radius suggests that the bulk concentration of mHttex1 particles is stable with time for cells with inclusions in the observed size range.

Growth of inclusion bodies formed in cells expressing mHttex1 (103q)-gfp was also examined; the results were very similar to the growth observed in mHttex1(72Q)-GFP-expressing cells (Supplementary Note 3 and Supplementary Fig. 11).

**The concentration of mHtt particles is proportional to that of soluble mHtt.** Cytoplasmic mHttex1-GFP has been shown to be largely soluble[9,10]. To better understand the kinetics of mHtt unfolding, particle formation, and inclusion body growth, we wanted to know the relationship between the cytoplasmic, soluble mHttex1-gfp concentration, and the concentration of unfolded mHtt particles that can be incorporated into the inclusion body.

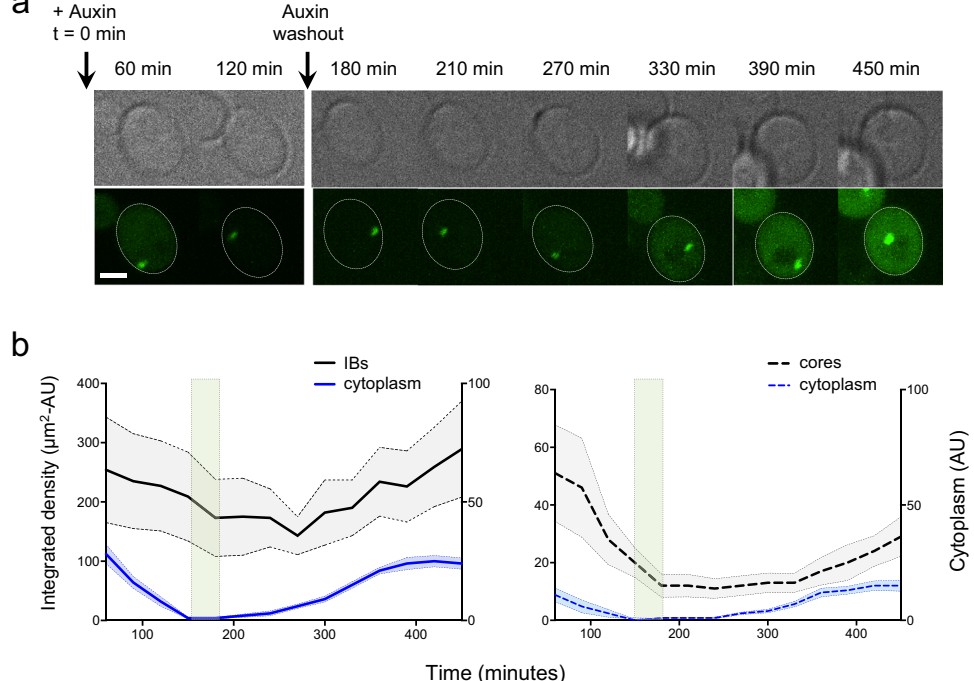

**Fig. 6 Persistent core particle of mHtt-GFP nucleates growth of a new inclusion body. a** Brightfield (upper) and a maximum projection of the green channel (lower) showing a cell containing a small mHttex1(72Q)-degron-GFP inclusion after the addition of auxin. Time after addition of auxin is indicated above the images, and the times of auxin addition and auxin washout are indicated by arrows. The outline of the cell containing the inclusion is outlined by a dashed white line. Scale bar, 2 μm. **b** Mean integrated density and cytoplasmic intensity vs time for ten inclusion bodies (IBs) and nine core particles (shaded regions indicate SEM). The shaded area between 150–180 min indicates the time of auxin washout.

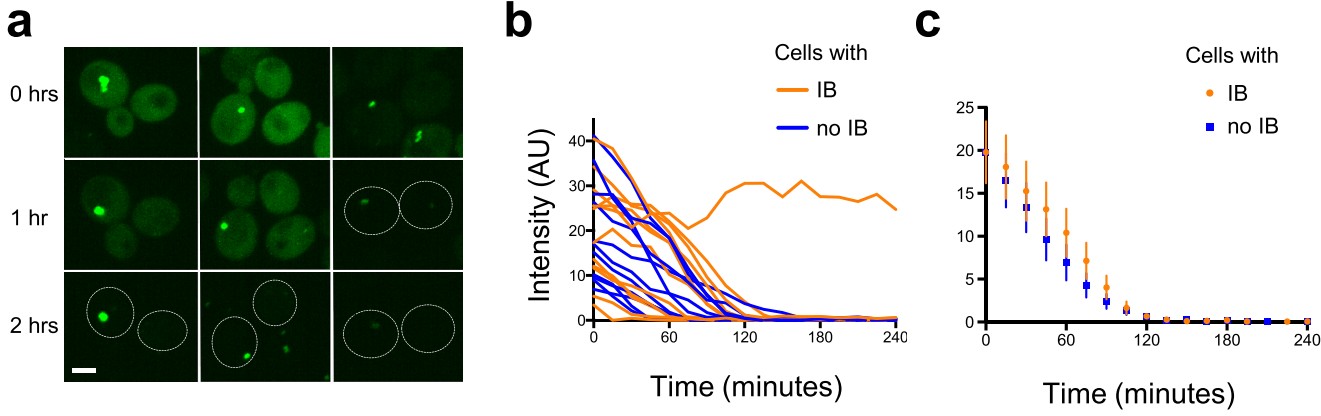

**Fig. 7 The proteasome has a similar capacity to acutely degrade cytoplasmic mHtt-GFP in cells with and without inclusion bodies. a** Time-lapse images of cells expressing mHttex1-degron-GFP were acquired every 15 min for 4 h after the addition of auxin. Maximum-intensity projection of three pairs of cells with similar cytoplasmic levels is shown; in the cell shown in the upper left image, the inclusion body (IB) was moving while the z-series was taken. Numbers indicate elapsed time. A confocal and brightfield image of the same field of cells is shown in Supplementary Fig. 9. Scale bar, 2 μm. **b** The cytoplasmic intensity of a random selection of cells with initial cytoplasm intensities between 5–45 AU with or without IBs, showing the loss of cytoplasmic mHttex1-degron-GFP after the addition of auxin at time 0. **c** The average cytoplasmic intensities of the cells shown in **b**, omitting the outlier. Error bars represent SEM; $n = 11$ for each category. Pairwise $t$-tests for each timepoint, without assuming a consistent SD and using the method of Benjamini, Krieger, and Yekutieli to correct for the false discovery rate, found no significant differences.

Unfortunately, we have not been able to directly visualize the bulk concentration of mHtt particles while simultaneously monitoring inclusion body growth. Imaging of mHttex1-GFP particles is possible but requires sustained excitation that significantly bleaches the cytoplasmic signal[8]. Therefore, we were not able to directly assay inclusion body growth and mHtt particle concentration in a single experiment.

However, the linear relationship between radius and time permitted us to assess the relationship between cytoplasmic

mHttex1-GFP concentration, and the underlying concentration of mHttex1-GFP particles that were added to the inclusion body. The data shown in Fig. 8a, b is consistent with the relationship:

$$\frac{dr}{dt} = kc_\infty \quad (4)$$

where $c_\infty$ is the bulk, cytoplasmic concentration of mHtt particles and $k$ is the reaction constant for the incorporation of mHtt particles into the inclusion body. We normalized and averaged

**Fig. 8 Inclusion body growth rate increases with cytoplasmic mHtt concentration and is proportional to the surface area. a** The measured volume of 25 inclusion bodies (IBs) over 3-5 h. The growth curves were aligned to the time of the first observation of each IB. The color of each IB trace indicates the average cytoplasmic intensity of the cell containing the IB (bar, right). **b** Average IB volume, area, and radius are plotted over time for the 16 IBs in cells with cytoplasmic intensities >1100 (error bars indicate SEM). Fits of the growth of volume with $t^3$ ($R^2 = 0.97$), an area with $t^2$ ($R^2 = 0.98$), and a radius with $t$ ($R^2 = 0.99$) are shown. If the growth of the radius with time is linear, it follows that the growth of area will grow as $t^2$, and volume will grow as $t^3$.

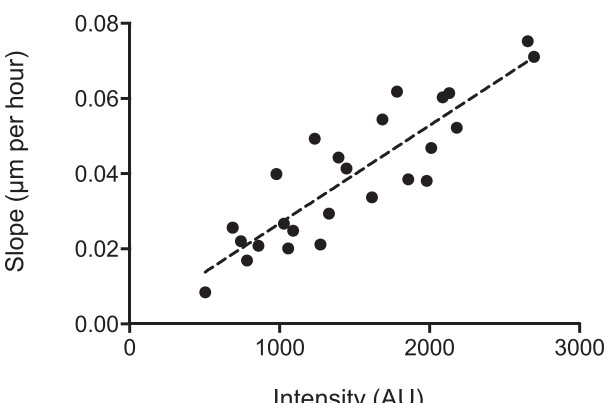

### Growth rate of IB radius vs cytoplasmic intensity

**Fig. 9 The concentration of mHtt particles is directly proportional to the concentration of soluble mHtt.** The growth rate of the radius of 25 individual inclusion bodies (IBs) (slope of the best-fit line) is plotted vs the intensity of the cytoplasm in which the IB was found. The distribution is fitted with a line ($R^2 = 0.77$).

the growth rate for 25 inclusion bodies coming from cells with cytoplasmic intensity ranging from ~500–2600 AU. By fitting the data for individual inclusions to a straight line, we obtained a slope equal to $k$ with an intercept at $r_0$.

For each inclusion body, we plotted radius vs time and fit the data with a straight line. Although these data are somewhat noisy, the coefficient of determination, $R^2$, was ≥0.85 for 19 out of 25 (Supplementary Table 2). The slope of the line gives the value of $kc_\infty$ for inclusion bodies in individual cells.

We then plotted $kc_\infty$ vs the cytoplasmic fluorescence, which is proportional to the concentration of soluble mHtt, for the cell containing that inclusion body (Fig. 9). The relationship appears linear. Values for slope ranged from 0.01–0.08 μm per hour, and the root-mean-square error for the fitted line was 0.009. These data suggest that the concentration of mHtt particles that are competent to fuse with the inclusion is directly proportional to the concentration of the soluble, cytoplasmic pool of mHttex1-GFP.

### Discussion

Our experiments provide insights into the mechanisms of inclusion body initiation and growth and allow us to create and test models of the dynamic partitioning of mHtt between a

cytoplasmic phase and an inclusion body phase. Expression of mHttex1-GFP from a low copy-number plasmid leads to natural variations in expression levels between cells of a population. We have further manipulated cellular mHtt concentrations using AID. Thus, we can use both natural and induced variation to probe the concentration dependence of inclusion formation and growth, and measure inclusion body growth and shrinkage at diverse cytoplasmic mHttex1-GFP levels.

Our results indicate that the formation of an inclusion body is initiated, possibly by nucleation, at a threshold cytoplasmic concentration and apparently regardless of proteasome capacity. The inclusion body grows by collision and absorption of small particles of aggregated protein, and the growth rate is dependent on the cytoplasmic mHttex1-GFP concentration. Inclusion bodies release material back into the cytoplasm, causing them to shrink when the concentration of cytoplasmic mHttex1-GFP drops to very low levels.

Direct observation of cells with diverse cytoplasmic mHttex1-GFP levels revealed a cytoplasmic steady-state threshold concentration below which cells never form inclusion bodies. Most (82%) of cells with mHttex1-GFP concentrations over this threshold form inclusions, but not all do, suggesting that other factors play a role in inclusion body formation, consistent with the known requirement for Hsp104 and Rnq1[9,31]. Conversely, in cells containing inclusion bodies where the cytoplasmic levels of mHttex1-gfp have been acutely reduced by auxin-mediated degradation, the existing inclusion body begins to shrink. Loss of material proceeds briskly until a small core particle or kernel, remains.

The presence of a persistent kernel of mHtt after the bulk of the inclusion body disappears indicates that there may be a non-phase-separated structure within the inclusion body. The over-whelming majority of inclusion body-forming cells form only one. After auxin is washed out and cytoplasmic mHttex1 rises, new material is added to the existing core particle, suggesting that it represents a nucleation site.

In solution, the presence of active centers can appreciably lower the energy barrier to nucleation of a phase-separated compartment; seeding may be used to speed the formation of phase-separated gels[32,33]. Nucleation is a mechanism by which the cell could control inclusion number and composition. Consistent with a nucleation model, we find that the time to form an inclusion body is not strongly dependent on cytoplasmic mHttex1-GFP concentration, unlike the growth of existing inclusions. This observation suggests that there is a regulated process that is initiated in cells with above-threshold levels of unfolded protein. Thereafter, the speed of growth of the inclusion body is dependent on the concentration of mHttex1-GFP.

A prevalent model of inclusion body formation states that inclusions form when the proteasome is overwhelmed, based on the increase in inclusion frequency seen when the UPS is inhibited[28–30]. Based on this model, we anticipated that cells with inclusions might not be able to carry out auxin-induced degradation because it acts through the UPS. However, we found that proteasomes can acutely degrade substantial quantities of mHttex1-degron-GFP in cells with and without inclusion bodies, suggesting that the UPS has the excess capacity both before and after they have formed. Taken together, these observations suggest that inclusion body formation is triggered by a regulated process when levels of unfolded protein reach a threshold, regardless of the capacity of the proteasomes. This is consistent with a model in which cells promote the formation of an inclusion body to prevent stress on the UPS[34].

Current models describe two fundamentally different mechanisms of growth for inclusions of mHttex1-GFP (Fig. 1a, b)[8,11,14,18]. In the phase separation model, the inclusion results from the diffusion and coalescence of small particles of aggregated material into a larger inclusion (Fig. 1a). Alternatively, material may be actively transported to inclusions, as shown in Fig. 1b. In the active transport model, an IPOD is generally considered to be docked at the vacuolar membrane while a substantial fraction of new material is transported to the inclusion body along actin cables, proposed to be mediated by Myo2 and/or Cvt vesicles[11,12,18,35]. In another version of the active transport model, the Sherman lab has proposed that the mHtt inclusion is an aggresome[14], located at the microtubule-organizing center and receiving material via retrograde transport along microtubules.

A key feature of the active transport models is that transport processes are limiting for inclusion growth. If transport is the limiting variable in inclusion growth, and the cytoplasmic concentration of material remains constant over the timescale of our experiment (consistent with our observations), then the volumetric growth due to the incorporation of new material into the inclusion body should be linear with time.

Alternatively, the phase separation model predicts that the inclusion body grows through the integration of particles of unfolded protein captured by its surface. This proposal is consistent with the observed mobility of the inclusion body and small particles and would be characteristic of other phase-separated compartments observed in cells. A model in which Httex1-GFP inclusion bodies incorporate small, diffusing particles of mHttex1-GFP would also account for the finding that other Hsp104-dependent inclusions have been shown to contain sub-particles of less soluble material[36–38].

If the inclusion body grows through collision with small particles of unfolded mHtt, the rate of volumetric inclusion of new material taken up by the inclusion should increase with increasing surface area, as the growing surface will result in a greater number of collisions. However, the incorporation of particles into the inclusion body may be limited either by the rate at which particles diffuse to its surface or by the speed of the incorporation reaction. If diffusion of particles to the surface is limiting, we would expect the cross-sectional area to grow linearly with time (1). If diffusion is not limiting, then we would expect the inclusion body radius to grow linearly with time (2).

Our measurements suggest that the inclusion radius does increase linearly over time, implying volumetric growth as $t^3$. In this way, the growth of the inclusion body is consistent with the previous modeling of the growth of nanoparticles in solution[39]. For the case where the surface reaction rather than diffusion of monomers is the limiting factor, then the change in radius with time has been described as

$$\frac{dr}{dt} = kv(C_b - C_r) \qquad (5)$$

where $r$ is the radius of the crystal, $k$ describes the rate of the surface reaction, $v$ is the molar volume of the monomer, $C_b$ is the concentration of the monomer in bulk solution, and $C_r$ is the solubility of the particle[39]. In our case, the cytoplasmic concentration of mHttex1-GFP is relatively constant for each cell during the 5–6 h timecourse. We assume, for moderate to large inclusions, that the surface concentration of mHttex1-GFP particles is also constant, which we believe to be reasonable for a large body whose surface is not undergoing large changes in curvature (surface tension).

Lastly, we conclude that the concentration of mHtt particles that are competent to fuse with the inclusion body is linearly proportional to the cytoplasmic, soluble mHttex1-GFP concentration. The growth of the inclusion body radius is constant with time, suggesting that the bulk concentration of mHtt particles, $c_\infty$, remains reasonably constant in individual cells, while $c_\infty$ increases linearly with the measured cytoplasmic intensity.

Formation of mHttex1-GFP particles is dependent on the dual-function aggregase/disaggregase Hsp104[8,40,41]. These findings suggest that substrate levels are not nearing saturation for Hsp104 in our cells.

The processes of inclusion formation and growth are highly dependent on cellular context. Indeed, in living *S. cerevisiae*, mHtt-GFP inclusions do not form in the absence of Hsp104 or Rnq1, whereas in vitro, mHttex1 forms amyloid fibrils and large aggregates spontaneously[26]. Strain background, Htt sequence, and polyQ tract length also modulate the types and frequencies of Htt inclusions[42]. It is not yet known whether all of these different mHtt inclusions or inclusions formed in other polyglutamine-expansion diseases, are also phase-separated or possess a core particle. Here, we focused on a detailed analysis of the kinetics and growth mechanism of the dominant form of inclusion body formed by mHttex1(72Q) in a standard wild-type yeast strain.

Our imaging studies, consistent with earlier biochemical studies, support a model in which the inclusion bodies are composed of discrete, insoluble particles that are absorbed into a larger phase-separated structure through collision, followed by coalescence. While consistent with the nucleation and growth of phase-separated compartments in general, our findings represent a significant change to the classical view of the unfolded protein deposit as an insoluble structure that receives additional material through active transport. A more accurate appreciation of inclusion structure and growth is necessary to understand their role in the processing and removal of unfolded protein from the cell and will allow us to address outstanding questions about inclusion toxicity.

## Materials and methods

**Strains, plasmids, and culture conditions**. Yeast strains were grown using standard conditions and the appropriate glucose-based selective media at 30 °C[43]. The lithium-acetate method was used for transformations; transformants were selected on the appropriate selective medium[44]. The doubling time of the mHttex1(72Q)-GFP strain in liquid culture was determined using cultures grown in SC-Leu at 30 °C with shaking. Overnight cultures were inoculated to $OD_{600} = 0.03$–$0.05$ and left to shake for 2 h, after which cells were in the exponential growth phase. The cultures were sampled approximately every hour for 4–5 h. Doubling time was calculated by plotting time vs $\ln(OD_{600})$, fitting with a straight line ($R^2 \geq 0.99$), and dividing the natural log of 2 by the slope.

Plasmids used in this study are listed in Supplementary Table 3. Mutant huntingtin expression plasmid pEB4 (mHttex1(72Q)-GFP) was constructed as described previously[8]; other Htt constructs were constructed in the same fashion. Briefly, the Htt-GFP sequences were amplified from Addgene Htt exon 1 plasmids 15582, 15833, and 1188 (Dr. Susan Lindquist). The sequences were subcloned into plasmid p415, a low copy CEN plasmid, under the control of the constitutive TDH3 promoter (more commonly known as the GPD, or glyceraldehyde-3-phosphate-dehydrogenase, promoter). The Htt sequences contain the complete HTT exon 1 coding sequence with the indicated number of glutamines. The ΔPRD construct (pEB6) removes the entire proline-rich domain after the first proline through the linker before GFP, a total of 49 amino acids.

The insertion of the IAA[71–114] degron sequence into pEB4 and pEB6 between the Httex1 and GFP was made using standard PCR-based subcloning methods; Kapa Hifi (KapaBiosystems) or Phusion DNA polymerase (New England Biolabs) was used for PCR reactions requiring high fidelity. All plasmid DNA sequences derived from a PCR amplicon were sequenced and found to be free of mutations that change coding or known regulatory sequences. Integration of Tir1 into the HO locus was carried out using a linearized plasmid[21].

Strains used in this study are listed in Supplementary Table 4. The parental strain for the strains used in this study was BY4741 *MATa his3Δ1 leu2Δ0 met15Δ0 ura3Δ0*.

**Confocal image acquisition and data analysis for inclusion body growth and cytoplasmic intensity studies**. Cells were inoculated in a selective medium and grown overnight at 30 °C with shaking; cultures in the mid-log phase the next day were selected for the imaging study. When performing time-lapse experiments, cells carrying pEB4 or pEB11 were mounted on a 2% agar pad made with selective media, covered with a #1.5 coverslip, and sealed with valap to prevent drying. Fields of cells were selected in brightfield mode and imaged every hour for 5–6 h. Individual images from the same stack shown together are displayed with consistent contrast settings.

Cells were imaged using a 100x/1.45 CFI Plan Apo Lambda objective lens on a TiE2-PFS microscope (Nikon) equipped with a CSU-X1 spinning-disk unit (Yokogawa Electric, Tokyo, Japan), a Zyla sCMOS camera (Andor, Belfast,

Northern Ireland), and OBIS LX 488 and LS 561 lasers (Coherent Inc, Santa Clara, CA).

Quantification was performed on unprocessed images using the Fiji distribution of ImageJ[45,46]. Using a script written by T.C.S. for ImageJ, the size and mean intensity of the inclusion body were measured in the optical section with the highest maximum intensity, assumed to be the section closest to the center of the inclusion. First, the mean cytoplasmic intensity was measured in a region excluding the vacuole and distant from the inclusion. After identifying the optical section with the highest maximum intensity, the inclusion was segmented using a threshold of 1.5x the mean cytoplasmic intensity. The Analyze Particles function was used to measure the area, position and mean, the minimum and maximum intensity of the inclusion. Mean cytoplasmic intensity was also reported.

Previous work has established that mutant Httex1(72Q)-GFP is synthesized and removed on a timescale that prevents correction for photobleaching in imaging studies conducted over a period of hours. At longer times, the population of molecules has been exposed to varying exposure times[8]. For this reason, excitations were widely spaced.

Mutant Httex1(103Q)-GFP-expressing cells were imaged using identical parameters to 72Q-expressing cells. In order to confirm that changes in laser power and alignment had not appreciably affected intensity values, mHttex1(72Q)-GFP-expressing cells were imaged immediately prior to the 103Q-expressing cells and the values for cytoplasmic intensity were compared to those obtained in the 72Q timecourse. Mean intensity for the 72Q-expressing cells in the timecourse was 1094 compared to 969 for the z-stacks preceding 103Q imaging ($n = 101$, 96). The overall range of cytoplasmic intensities for the 72Q timecourse was 45 to 3194, compared to a range of 177 to 2859 for the z-stacks.

**Analysis of inclusion integrated density and size**. ID was calculated as the mean inclusion intensity multiplied by the area of the inclusion; mean inclusion intensity was corrected for background intensity. Inclusions are close to circular[8] and were approximated as spherical for the purpose of calculating diameter, area, and volume.

For each inclusion, the area through the plane of maximum intensity was taken as the maximum area for each inclusion. Radius was calculated as $(area/\pi)^{1/2}$, and volume was calculated as $4/3 \times area \times radius$ ($4/3\pi r^3$). Calculations were done in Excel (Microsoft Office), Prism (GraphPad, San Diego), and R 3.6.2 (R Core Team, Vienna, Austria, 2020; https://www.R-project.org). Prism and R were used for graphing functions and statistics. Line fitting in Prism and R was used to determine the slope, standard error, 95% confidence interval, and value for $R^2$ of the best-fit line for inclusion body radius vs time.

**Auxin treatment and image analysis**. MatTek dishes (MatTek Corporation) were prepared by adding ~100 μL of a filter-sterilized 2 mg/ml solution of concanavalin A, allowing them to stand for 30 min at room temperature, rinsing with distilled water and allowing them to dry overnight[47]. One hundred microliters of an overnight culture of log phase cells were added to the MatTek dish and allowed to settle for 10 min before being rinsed gently with SC-leu. After rinsing, 200 μL of SC-leu was added to the dish. Fields of cells were selected in brightfield. Prior to the addition of auxin, two fields of cells were imaged ten times as rapidly as possible to measure photobleaching, using the same settings that were used in the imaging experiment.

A fresh solution of naphthalene acetic acid (Sigma-Aldrich) was prepared to 60 mM in 95% ethanol and diluted to 2.7 mM in water. Twenty microliters of 2.7 mM naphthalene acetic acid (NAA) was added to the center of the well, for a final concentration of 250 μM. Eight new fields of cells were imaged every 15 min for 4 h. For vehicle controls, 20 μL of 95% ethanol was added to the cells instead of NAA.

To correct for photobleaching, using the images collected before auxin treatment, five cells without an inclusion were selected and a region of interest was drawn to avoid the vacuole. The mean cytoplasmic intensity in cells was measured over ten imaging cycles. The intensity measurements were normalized to the initial intensity, averaged, and a line was fitted using an exponential decay function. The slope of the line and number of rounds of imaging were used to determine the fraction of fluorescence loss due to photobleaching and correct mean intensity in auxin-treated cells.

Using the ImageJ script described above under confocal image acquisition and data analysis, the size and mean intensity of the inclusion body in the optical section with the highest intensity was determined. Intensities were corrected for photobleaching, as the population of inclusion bodies used in this study ceased incorporation of mHttex1-GFP soon after the onset of imaging and therefore contained a stable or declining population of molecules that one could reasonably assume were subjected to the same, or very close to the same total number of excitations. Integrated densities and volumes were calculated as described above.

In our analysis of the loss of fluorescence in moderate to large inclusion bodies, a Fiji plugin was used to automate measurements of inclusion size. In the analysis of the inclusion body core particle, however, the particle was identified manually and a region of interest was drawn around it. Kernels were defined as particles with an ID $\leq 25 \, \mu m^2$ AU. Kernels that were too small to be detected using a threshold of 1.5x the cytoplasmic intensity were thresholded by eye to 1.1–1.2x above the cytoplasm using the Threshold function in ImageJ. The ImageJ Analyze Particles function was used to measure mean intensity and cross-sectional area. The cytoplasmic background was subtracted. The images were not corrected for photobleaching as the Httex1-GFP was newly and continuously synthesized during the recovery period.

For studies of mHttex1ΔPRD-degron-GFP, the NAA was tested on mHttex1-degron-GFP-expressing cells to confirm its potency.

**Auxin washout**. NAA was added to mid-log cultures to a concentration of 250 μM and incubated at 30 °C with shaking for 45 min. The cells were seeded onto ConA-treated MatTek dishes, the excess medium and unattached cells removed and fresh medium with 250 μM NAA added to 500 μL. The cells were imaged every 30 min for 2 h, and the NAA was washed out by dilution using five washes with 5 mL pre-warmed medium. Each wash was carefully removed to leave about 250 μL, resulting in a 20-fold dilution with each wash, or an overall dilution of ~1.6 × 10⁶–fold. Imaging was restarted 30 min after the previous timepoint and the recovery was imaged every 30 min for 5 h.

### Inclusion growth rate calculations

*Diffusion and collision-driven process.* If we assume that the inclusion body grows as a result of the incorporation of diffusing aggregative particles, we can calculate the mass balance on the inclusion body as follows.

The mass $m$ of the inclusion body increases due to the flux of aggregative particles to the surface. We assume the origin of our coordinates to be at the center of the inclusion body, taken as a sphere:

$$\frac{dm}{dt} = -\int \mathbf{j}_m \cdot \hat{n}\, dS(t) = -\int j_{m,r}\, dS(t) \tag{1}$$

where $j_{m,r}$ is the radial component of the mass flux $\mathbf{j}_m$, and $\hat{n}$ is the outward-facing unit normal from the surface of the inclusion body. As it is evident that the inclusion body grows, we can say that material is moving toward the origin, or $\mathbf{j}_m \cdot \hat{n} = j_{m,r} < 0$. The mass is $m = \rho V$, and assuming that the inclusion body has a constant density, $\rho$, and spherical symmetry, this equation can be written for volume growth:

$$\rho \frac{dV}{dt} = -\int j_{m,r}\, dS(t) = -4\pi r(t)^2 j_{m,r} \tag{2}$$

For the assumption of a spherical inclusion body,

$$\frac{dV}{dt} = 4\pi r(t)^2 \frac{dr}{dt} \tag{3}$$

and (2) can be rewritten as

$$\frac{dr}{dt} = -\frac{j_{m,r}}{\rho} \tag{4}$$

The flux $j_{m,r}$ can be written in volumetric form $j_r$ by using the density of the aggregative particles, i.e., $j_{m,r} = \rho_{\text{part}}$, to yield

$$\frac{dr}{dt} = -\frac{\rho_{\text{part}}}{\rho} j_r \tag{5}$$

where it can be recalled that $\rho = \rho_{\text{IB}}$. It is not clear that the density of the aggregative particles are the same as the density of the inclusion body, which appears to have a heterogeneous structure. However, if we take $\rho_{\text{part}}/\rho$ as equal to a constant $\kappa_{\text{den}}$, we can say

$$\frac{dr}{dt} = -\kappa_{\text{den}} j_r \text{ or } r = r_0 + \kappa_{\text{den}} j_r t \tag{6}$$

If we assume a steady-state concentration develops around the inclusion body, we can say that the concentration must satisfy the diffusion equation:

$$\frac{1}{R^2} \frac{d}{dR}\left(R^2 \frac{dc}{dR}\right) = 0 \tag{7}$$

The boundary conditions are $c(R \to \infty) = c_\infty$, and the surface flux at $R = r$, where $r = r(t)$:

$$-j_r = D \frac{dc}{dR}\bigg|_{R=r} = kc(r), \tag{8}$$

where $D$ is the diffusivity of the aggregative particles in the cytoplasm, and $k$ is the rate constant described above. The solution is $c = A + B/R$ with $A = c_\infty$ and $B$ is determined from the flux balance to yield

$$c(R) = c_\infty\left(1 - \frac{1}{Z+1}\frac{r(t)}{R}\right), Z = \frac{D}{kr} \tag{9}$$

where $Z$ may be considered as a dimensionless diffusivity and is the inverse of the commonly used Damköhler number, $Da = 1/Z = kr/D$.

For $Z \ll 1$ (i.e., $Da \gg 1$), the reaction rate is fast and limiting cytosol concentration as the surface is approached vanishes, $c(r(t)) \to 0$. In this case, we may rewrite (8) as

$$c(R) = c_\infty\left(1 - \frac{r(t)}{R}\right), \tag{10}$$

and the flux is

$$j_r = -D \frac{dc}{dR}\bigg|_{R=r} = D\frac{c_\infty}{r(t)} \tag{11}$$

For fast reaction rates, we can say

$$\frac{dr}{dt} = -\kappa_{\text{den}} D \frac{c_\infty}{r(t)} t \text{ or } r^2 = r_0^2 + 2kc_\infty t \tag{12}$$

In this case, the surface area is predicted to grow linearly with time.

If the reaction rate is slow, $Z \gg 1$ (or $Da \ll 1$), and the combination of aggregate diffusion and movement of the inclusion body has time to make the concentration field uniform (i.e., these factors prevent a stable concentration gradient from forming around the inclusion), then the concentration of particles at the inclusion body surface, $c(r(t))$, will asymptotically approach the bulk concentration of particles in the cytosol, $c_\infty$. In this case, we may say that

$$r = r_0 + kc_\infty t \text{ or } \frac{dr}{dt} = kc_\infty \tag{13}$$

where $k$ is the forward reaction constant for the integration of particles into the inclusion body.

*Transport-driven process.* If we assume that material is transported to the inclusion body by active transport systems, then the rate of material reaching the inclusion is dependent on, and limited by, the carrying capacity of the transport systems. If we assume that the volume of material carried to the inclusion body is constant with time, then we can say

$$\frac{dV}{dt} = k_{\text{tr}} \tag{14}$$

where $k_{\text{tr}}$ is the rate constant for the transport systems delivering material to the inclusion body in the volume of material per unit time. If the capacity of the transport system is limiting, at some time after initiation of the inclusion body, we can say

$$V(t) = k_{\text{tr}} t \tag{15}$$

Substituting $4/3\, \pi r(t)^3$ for volume, we can say

$$\frac{4}{3}\pi r(t)^3 = k_{\text{tr}} t, \tag{16}$$

which can be rearranged to give:

$$r(t) = \left[\frac{3k_{\text{tr}}}{4\pi}\right]^{\frac{1}{3}} t^{1/3} \tag{17}$$

Therefore,

$$\frac{dr}{dt} = \frac{1}{3}\left[\frac{3k_{\text{tr}}}{4\pi}\right]^{\frac{1}{3}} t^{-2/3} \tag{18}$$

in the case where the material is transported to the inclusion body by transport systems with a capacity that is independent of the size of the inclusion.

**Statistics and reproducibility**. Growth curve fits and $R^2$ values were obtained using the lm function (fit linear models) in R 3.62. Other statistical analyses were performed in Microsoft Excel or GraphPad Prism 7. Statistical significance was determined using Student's *t*-test for normally distributed data, and either the Mann–Whitney or the Kruskal–Wallis test for non-normally distributed data. Normality was determined by inspection or by the D'Agostino-Pearson test in Prism.

**Reporting Summary**. Further information on research design is available in the Nature Research Reporting Summary linked to this article.

## Data availability

All data generated and analyzed for this study are included in the article and in Supplementary Data 1. Additional raw data (images, etc.) are available upon reasonable request from the authors.

## Code availability

All scripts used in this study are available on Zenodo at https://doi.org/10.5281/zenodo.5082513[48]. Scripts were run in Fiji 2.1.0/1.53c and R 3.62.

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

## Acknowledgements

The authors thank Liza Pon, Istvan Boldogh, Janeska de Jonge, and Enrique Garcia for their valuable advice and assistance with strain construction. We are grateful to Peter Lipke for his advice on the manuscript. We would also like to thank Pat Hooper for the helpful discussion. This work was supported by grants from the National Institutes of Health to L.E. (SC2GM116697 and SC3GM136517). Images were collected on a Nikon Spinning Disk microscope acquired by Department of Defense Equipment Award W911NF-17-1-0516.

## Author contributions

L.E. conceived the ideas, performed the experiments and analyses, and wrote the manuscript. S.P. made strains and performed experiments. J.F.M. derived analyses of growth. T.C.S. contributed to image and data analysis.

## Competing interests

The authors declare no competing interests.
