## [Transparent Peer Review File · Communications Biology]

Reviewers' comments:

Reviewer #1 (Remarks to the Author):

The manuscript by Pei et al. report a valuable work on the formation of Huntington inclusions in cells. The presentation and discussion are clear, the observations are well taken and the analysis is statistical robust and experimentally solid. It is an important effort to drive a quantitative and theoretical analysis after biological observations. I have only some minor comments.

1. While the parameter "Integrated density" is defined in the Method section, it suddenly appears in the main discussion with no clear meaning. Moreover, its significance and convenience with respect to other quantities should be pointed out.

2. Cluster-like inclusions are explained, but they could be made clearer by showing an image (particularly in manuscript that is based on visual/microscopy assays).

3. Which is the nature and origin of the core? Is it a more energetically stable nucleus? Or is it the hallmark of two distinct coagulation processes? Or is it an aging product occurring in the crowded nucleus after coagulation? The answer would be quite interesting. Whether this is presumably beyond the scope of this work, the authors may want to discuss it, even if in a dubitative way.

Reviewer #2 (Remarks to the Author):

Pei et al – "Threshold concentration and random collision determine the growth of the huntingtin inclusion from a stable core". referee report

Pei and colleagues present a very compelling set of data exploring the mechanisms behind aggregation of mutant huntingtin in yeast cells. The authors argue and provide evidence to show that aggregation/amyloid formation is initiated by a random collision of mutant huntingtin into a phase-separated condensate composed of a particle-rich mixture. This process is merely concentration dependent and not related to an active transport as has been speculated from its accumulation in so-called iPODs. While the condensates are gel-like and reversible upon a decrease in overall concentration by a ubiquitin-dependent, proteasomal degradation, they can contain a small inner core (amyloid?) that is immobile and less or not reversible when the overall concentration is lowered.

Overall, the manuscript is well written, the methodology is clear and nicely explained. The chosen microscopy approach provides interesting observations, even though it is limited in some respects. Nevertheless, it is my opinion that the work should be accepted for publication, provided the authors address the issues discussed below.

Major comments:

1. Due to the important differences in aggregation kinetics of Htt variants either encompassing the full exon 1, or deletion mutants (such as deletion of the first 17 amino acids or the prolinerich region), as reported by several groups, the manuscript should clearly state whether a complete exon 1 fragment of Htt was used. In fact, it would be important to compare a few of these fragments to test to what extent the condensation step depends on the polyQ flanking regions or whether concentration dependent condensations also occurs for pure polyQ stretches only. This is also of importance for the putative relevance of these findings to other polyQ-containing disease causing proteins.

2. Regarding the mHtt(72Q)-degron-GFP construct, do authors know how it behaves in cells that do

not express the Ub ligase Tir1? Does it form IBs in a similar percentage of cells, and at the same rate, compared to mHtt(72Q)-GFP?

3. In figure 3c, the variances of cytoplasmic intensities between cells without and with IBs seems to be very different. Was that taken into account when performing the unpaired t test?

4. What statistical procedure was used to assess the differences between curves reported on figure 6C?

5. It might not be immediately clear to those unfamiliar with the mathematical aspect of the work what the authors mean by "t, t₂, and t₃" in the legend of figure 7. However, this is nicely explained in the legend of table S1. I suggest replacing and adapting the sentence "if the growth of the radius with time is linear, it follows that the growth of area will grow as t², and volume will grow as t³." from table S1 to the legend of figure 7.

6. I agree that the treatment of IBs as perfect spheres is a very adequate generalization for mathematical modeling and fitting of their data, and the modelling of more complex topologies would likely be prohibitive. However, while IBs indeed are mostly round under conventional microscopy, others have shown by cryo-EM methods that IBs can adopt more amorphous conformations that substantially deviate from a sphere (see for instance the many protrusions in IBs detected in yeast cells by Gruber et al., 2018 [DOI:10.1073/pnas.1717978115] and in mammalian cells by Bäuerlein et al., 2017 [DOI:10.1016/j.cell.2017.08.009]). Considering the aggregation mechanism proposed by the manuscript, perhaps authors could briefly discuss whether their modelling strategy could be influenced/skewed by such topologies, which likely have a much larger contact surface than a spherical IB. Perhaps in these cases the rate of soluble huntingtin accretion to the IB would not be linear?

7. In the discussion section the authors mention the crucial role of Hsp104 in IB formation/clearance. Indeed, it would be extremely interesting to show whether the shrinkage of IBs, as reported on figure 4, is impaired in the background of Hsp104 mutations/deletion. Data from such strains might provide invaluable mechanistic detail, which could lend more support to the authors' observations.

8. I missed a more thorough contextualization and generalization of the authors' findings in the discussion section regarding other recent papers that have shown complementary findings. For instance, the in vitro data from Posey and colleagues (2018; DOI:10.1074/jbc.RA117.000357) also suggests there is a concentration saturation for the aggregation of huntingtin, which is in line with the authors' findings that cells with GFP levels below a certain intensity rarely develop IBs. That deserves mentioning and referencing.

Similarly, the work of Peskett and colleagues (2018; DOI: 10.1016/j.molcel.2018.04.007) should be mentioned (as discussed by Aktar et al., 2019), especially since it is becoming clearer that there are major phenotypic differences in IB formation depending on yeast strain that is used.

Minor textual comments:

1. Line 62: "... protein using Htt fused GFP from the constitutive...". Perhaps "GFP-fused Htt", or "Htt fused to GFP"?

2. Line 96: "... appears to be a threshold concentration...".

3. Line 484: "... (consistent with our observations) ...". Extra space after the closing parenthesis.

4. Line 517: "...concentration of mHtt particles that are that are competent...".

5. Line 545: "...after which cells were reproducibly in exponential growth phase.". Perhaps there is a verb missing before "reproducibly"?
6. Line 664: "...that the IB has a constant mass density...".
7. Lines 700-701: "...concentration as the the surface...".
8. Lines 843-844: part of the title of reference 29 is in uppercase letters.

Reviewer #3 (Remarks to the Author):

Pei and colleagues set a study to understand in more details how mHtt inclusions are formed in living cell. They chose yeast and mHtt(72Q)-GFP as a model for their study. Using series of microscopic imaging they tracked formation and degradation of mHtt inclusions in living yeast cell. Basing on their data they calculated that new mHtt species seems to be incorporated into the inclusion by random collision and coalescence, rather than active transport as was previously described for the IPOD.

Manuscript is well written with interesting observations. Data generated in this study will be interesting for the community. The experiments are well planned with sufficient explanation in the text. However, there are few points which need to be clarified.

Comments:

1. Authors have shown that there is a certain threshold for cytoplasmic concentration of mHtt(72Q)-GFP from which IB are formed. Does this correlate with the length of the expressed polyQ region? Authors should compare threshold concentrations of polyQ(72) with longer version of the polyQ expansion.
2. Figure 4: Authors should include control were proteasome activity or autophagy pathway is blocked to confirm the degradation pathway for mHtt-GFP.
3. Figure 4: Would inhibition of protein translation in this experiment enhance the clearance of IB?
4. Figure S4: The conclusions should be milder as the number of cells analysed is not satisfactory (n=4). Did the authors followed the CLIs over longer time to see if they eventually form a round, IB like structures?
5. Figure 5: Authors observed that the core particle of IBs stays stable in the cell. What will happen when auxin is washed away? Is the core particle growing back, or is there another IB forming, distinct from the original IB?
6. Figure 6: controls should be included showing that the drop in the intensity (for both cells with or without IB) is prevented by the presence of proteasome inhibitor and not by inhibition of autophagy.
7. Figure 7: Does increase in the length of the polyQ expansion increases the growth rate of IBs? Is the growth rate of IBs changed in the condition when cellular active transport is impaired?

Minor comment:

1. In discussion part in lane 441-442 authors suggest that other factors might play a role in IB formation. Could they expand this with some examples?

2. Lane 517: spelling error- 'that are' is used twice.
3. Figure S2B: fix the graph to have 5h time point.

Dear Editors,

We thank the editors and reviewers for their time and critical feedback. Their suggestions have improved our manuscript and we're very grateful for their input. The particulars are addressed at length below, and in the revised manuscript as noted.

Response to Reviewers

Reviewer #1 (Remarks to the Author):

The manuscript by Pei et al. report a valuable work on the formation of Huntington inclusions in cells. The presentation and discussion are clear, the observations are well taken and the analysis is statistical robust and experimentally solid. It is an important effort to drive a quantitative and theoretical analysis after biological observations. I have only some minor comments.

- 1. While the parameter "Integrated density" is defined in the Method section, it suddenly appears in the main discussion with no clear meaning. Moreover, its significance and convenience with respect to other quantities should be pointed out.*

Author Response: Thank you for pointing out this omission. We have added the definition of integrated density to the main body of the text.

- 2. Cluster-like inclusions are explained, but they could be made clearer by showing an image (particularly in manuscript that is based on visual/microscopy assays).*

Author Response: An image of a CLI has now been incorporated into Supplementary Figure 3, which is a newly added figure that characterizes the frequency of inclusion types in mHtt(103Q)-GFP-expressing cells.

- 3. Which is the nature and origin of the core? Is it a more energetically stable nucleous? Or is it the hallmark of two distinct coagulation processes? Or is an an aging product occurring in the crowded nucleus after coagulation? The answer would be quite interesting. Wjheter this is presumably beyond the scope of this work, the authors may want to discuss it, even if in a dubitative way.*

Author Response: Studies of phase separation in vitro show that seeds can differ in composition from the larger structures that they nucleate. In mHtt-expressing cells the composition of the persistent core particle is unknown; however it is conceivable that it includes a more energetically stable form of mHtt, and it may contain other proteins as well. Currently, two proteins are known to be required for mHtt inclusions to form: the aggregase/disaggregase Hsp104, and the low sequence complexity protein Rnq1. We do not yet know whether Hsp104 or Rnq1 are required for the initiation of inclusions or for their ongoing growth, or both; if the former is true, it is possible that these proteins may be part of the core.

Reviewer #2 (Remarks to the Author):

Pei et al – “Threshold concentration and random collision determine the growth of the huntingtin inclusion from a stable core”. referee report

Pei and colleagues present a very compelling set of data exploring the mechanisms behind aggregation of mutant huntingtin in yeast cells. The authors argue and provide evidence to show that aggregation/amyloid formation is initiated by a random collision of mutant huntingtin into a phase-separated condensate composed of a particle-rich mixture. This process is merely concentration dependent and not related to an active transport as has been speculated from its accumulation in so-called iPODs. While the condensates are gel-like and reversible upon a decrease in overall concentration by a ubiquitin-dependent, proteasomal degradation, they can contain a small inner core (amyloid?) that is immobile and less or not reversible when the overall concentration is lowered.

Overall, the manuscript is well written, the methodology is clear and nicely explained. The chosen microscopy approach provides interesting observations, even though it is limited in some respects. Nevertheless, it is my opinion that the work should be accepted for publication, provided the authors address the issues discussed below.

Major comments:

- 1. Due to the important differences in aggregation kinetics of Htt variants either encompassing the full exon 1, or deletion mutants (such as deletion of the first 17 amino acids or the prolinerich region), as reported by several groups, the manuscript should clearly state whether a complete exon 1 fragment of Htt was used.*

Author Response: Our expression constructs contain the complete HTT exon 1, as we've now clarified in the Methods section (lines 596-603). We've additionally incorporated the common designation 'Httex1' into the MS to be clear about the construct used. The sequence of our mHttex1(72Q) construct is identical to that the HTT exon 1 sequence found in the NCBI Nucleotide database except for the length of the polyQ tract [Homo sapiens huntingtin (HTT), RefSeqGene (LRG_763) on chromosome 4].

In fact, it would be important to compare a few of these fragments to test to what extent the condensation step depends on the polyQ flanking regions or whether concentration dependent condensations also occurs for pure polyQ stretches only. This is also of importance for the putative relevance of these findings to other polyQ-containing disease causing proteins.

Author Response: The stability of pure polyQ tracts has been studied in solution: polyQ tracts become increasingly aggregative as the number of Qs increases (Chen et al., 2001; Walters and Murphy, 2009). However, pure polyQ peptides become aggregative at polyQ tract lengths well below what is observed in the context of HTT; pure polyQ peptides as short as 24Q have been shown to be aggregative whereas Httex1(30Q) has

been shown to be soluble in vitro (Scherzinger et al., 1997). In short, in the absence of flanking sequences to create and stabilize structure, the aggregative tendencies of pure polyQ are shifted towards shorter polyQ tracts. We also note, however, that polyQ tracts in disease-causing proteins are flanked by other sequences and polyQ tracts do not naturally occur in isolation.

Httex1 contains an alpha helix, the polyQ tract and a proline-rich domain (PRD); the structure of the PRD and its effect on polyQ structure is the subject of debate. While it has already been shown that Httex1 Δ PRD-GFP is more highly aggregative (Dehay and Bertolotti, 2006), we have examined the contribution of the proline-rich sequence flanking the polyQ tract to inclusion structure in vivo.

We first attempted to image growth of the PRD deletion (Δ PRD) constructs by doing a timecourse of cells expressing Httex1(72Q) Δ PRD-GFP. However, deletion of the PRD destabilized the protein to such an extent that there was very little cytoplasmic protein visible and cells did not form large singular inclusions but instead cells were born with numerous small inclusions distributed throughout the cell, which were below the limit of our resolution. Thus, we were unable to relate cytoplasmic intensity to inclusion initiation and growth for the Δ PRD construct.

Therefore, we used AID to probe the relationship between cytoplasmic protein and Δ PRD inclusions by introducing the degron into the Δ PRD construct. Interestingly, the addition of a degron in between Htt and GFP appeared to stabilize the protein enough for it to form larger, though still abnormal, inclusions. We found that large inclusions formed by the Httex1(72Q) Δ PRD-degron-GFP construct are insensitive to loss of cytoplasmic Httex1 Δ PRD (i.e., they don't shrink). We concluded that the Httex1 Δ PRD inclusions are not phase separated, don't release material back into the cytoplasm, and no concentration dependence was observed by this assay. These findings have been added to the Results and Discussion sections (lines 257-263, Supplementary Note 2, Supplementary Fig. 8).

2. *Regarding the mHtt(72Q)-degron-GFP construct, do authors know how it behaves in cells that do not express the Ub ligase Tir1? Does it form IBs in a similar percentage of cells, and at the same rate, compared to mHtt(72Q)-GFP?*

Author Response: Thank you for pointing out this omission. We compared the expression of mHttex1(72Q)-GFP and mHtt(72Q)-degron-GFP in cells lacking Tir1, and incorporated the results into the MS (lines 213-216). Briefly, the IBs are very similar in appearance and frequency to those in cells expressing mHtt without the degron. We have been using a short version of the degron tag containing amino acids 71-114; it does not appear to significantly affect mHtt stability or properties associated with IB formation.

3. *In figure 3c, the variances of cytoplasmic intensities between cells without and with IBs seems to be very different. Was that taken into account when performing the unpaired t test?*

Author Response: We thank the Reviewer for bringing this to our attention. We've redone the t-test assuming unequal variance and noted it in the figure legend.

4. *What statistical procedure was used to assess the differences between curves reported on figure 6C?*

Author Response: The points were compared using pairwise t-tests for each timepoint, without assuming a consistent SD, and using the method of Benjamini, Krieger and Yekutieli to correct for the false discovery rate. No significant differences were found. A description of the analysis has been added to the figure legend.

5. *It might not be immediately clear to those unfamiliar with the mathematical aspect of the work what the authors mean by "t, t2, and t3" in the legend of figure 7. However, this is nicely explained in the legend of table S1. I suggest replacing and adapting the sentence "if the growth of the radius with time is linear, it follows that the growth of area will grow as t2, and volume will grow as t3." from table S1 to the legend of figure 7.*

Author Response: This clarifying text was added to the figure legend (now Figure 8).

6. *I agree that the treatment of IBs as perfect spheres is a very adequate generalization for mathematical modeling and fitting of their data, and the modelling of more complex topologies would likely be prohibitive. However, while IBs indeed are mostly round under conventional microscopy, others have shown by cryo-EM methods that IBs can adopt more amorphous conformations that substantially deviate from a sphere (see for instance the many protrusions in IBs detected in yeast cells by Gruber et al., 2018 [DOI:10.1073/pnas.1717978115] and in mammalian cells by Bäuerlein et al., 2017 [DOI:10.1016/j.cell.2017.08.009]). Considering the aggregation mechanism proposed by the manuscript, perhaps authors could briefly discuss whether their modelling strategy could be influenced/skewed by such topologies, which likely have a much larger contact surface than a spherical IB. Perhaps in these cases the rate of soluble huntingtin accretion to the IB would not be linear?*

Author Response: With regard to the question of surface topology, our results do not rule out the possibility of a more complex surface. Even assuming convoluted topology, the surface area will remain $\propto r^2$. For absorption following a collision, as long as the surface area is $\propto r^2$ the rate of change with time should be the same, although the rate constant would be proportionally larger with larger surface area.

With regard to the study by Gruber and colleagues, we find that about 1/7th of the inclusions in mid-log mHtt(103Q)-GFP-expressing cells are CLIs (lines 196-200, Supplementary Note 1, Supplementary Fig. 3), and the percentage of inclusions that are CLIs goes up substantially when the cells are allowed to exit log phase, for example, in saturated overnight cultures. Gruber and colleagues report only minimal observations of inclusions by fluorescence microscopy in living cells, nor do they describe their culture conditions. Furthermore, Gruber and colleagues see no internal structure at all

in inclusions formed by either mHttex1(97Q) or mHttex1(97Q) Δ PRD by cryo-ET. However, inclusions formed by mHttex1 with and without the PRD look and behave very differently in living cells (lines 257-263, Supplementary Note 2, Supplementary Fig. 8), suggesting that there are significant differences in structure.

The example of a mHttex1(97Q) IB shown in Gruber et al. is ovoid, although the surface is not perfectly smooth. There are a number of possible interpretations, including: (1) they have shown a CLI rather than an IB; (2) the outer surface of IBs is not perfectly smooth; or (3), any fixation method will introduce some artifacts, so this may be a result of the fixation process. We have no way of distinguishing between these possibilities at present.

Fibers were seen by both Gruber et al and Bauerlein et al, but the larger mammalian IBs seen in the cryo-ET by Bauerlein and colleagues contain clear internal detail of fibers. Bauerlein and colleagues assume that the fibers in the inclusions are amyloid fibers composed of Htt, which is reasonable. The cryo-ET images suggest that the fibers radiate outward from the center, consistent with a site of nucleation. However, there is abundant 'empty space' in their model of radiating fibers, which could be filled with non-amyloid material not visible by ET.

Based on the diffusion constant of small Htt particles, we have proposed that they could contain amyloid fibers of Htt, as well as Hsp104 and possibly other associated proteins (Aktar et al., 2019). As small particles move randomly, we are proposing that they fuse with the IB after collision. Based on the fluorescence recovery showing material incorporation throughout IBs, it appears that there is internal redistribution of contents in yeast mHtt IBs. The consistently ovoid appearance and other characteristics associated with phase separation suggests surface tension at the interface between IB and cytoplasm, however it remains possible that the surface is not smooth. The features that we have observed in vivo are not *inconsistent* with the appearance of the inclusion seen in Gruber et al, or by Bauerlein and colleagues. However, the absence of visible internal features in existing cryo-ET of yeast IBs prevents us from drawing any definitive conclusions about structure from the cryo-ET at this time.

7. *In the discussion section the authors mention the crucial role of Hsp104 in IB formation/clearance. Indeed, it would be extremely interesting to show whether the shrinkage of IBs, as reported on figure 4, is impaired in the background of Hsp104 mutations/deletion. Data from such strains might provide invaluable mechanistic detail, which could lend more support to the authors' observations.*

Author Response: We agree that these are critical experiments. Hsp104 is absolutely required for inclusions to form, making it difficult to probe the possible role of Hsp104 in disaggregation. We are trying several approaches to convincingly separate the aggregate and disaggregase functions of Hsp104, but feel that these studies are outside the scope of this MS.

8. *I missed a more thorough contextualization and generalization of the authors' findings in the discussion section regarding other recent papers that have shown complementary findings. For instance, the in vitro data from Posey and colleagues (2018; DOI:10.1074/jbc.RA117.000357) also suggests there is a concentration saturation for the aggregation of huntingtin, which is in line with the authors' findings that cells with GFP levels below a certain intensity rarely develop IBs. That deserves mentioning and referencing. Similarly, the work of Peskett and colleagues (2018; DOI: 10.1016/j.molcel.2018.04.007) should be mentioned (as discussed by Aktar et al., 2019), especially since it is becoming clearer that there are major phenotypic differences in IB formation depending on yeast strain that is used.*

Author Response: The paper by Posey and colleagues is very interesting with regard to HTT phase transitions in vitro, and we thank the reviewer for the reference.

With regard to profilin, *S. cerevisiae* do have a profilin isoform (Pfy1), so is conceivable that the differences in aggregation between mHtt with and without the proline-rich domain could be due in part to a profilin-mediated modulation of the phase transition. As profilin has been shown to bind to poly-L-proline (Krishnan and Moens, 2009; Mahoney et al., 1997), such as found in HTT exon 1, it is possible, for example, that our mHttex1-GFP construct is stabilized by profilin binding to the polyP sequence directly after the polyQ tract. Our data suggests that inherent protein stability is one determinant of which aggregation pathway will be followed by a protein in vivo.

More generally, we agree that there are strain-to-strain variations in inclusion formation, as many proteins besides Htt modulate this process in vivo. We've raised these points in the Discussion (lines 563-569).

Minor textual comments:

1. Line 62: "... protein using Htt fused GFP from the constitutive...". Perhaps "GFP-fused Htt", or "Htt fused to GFP"? Corrected
2. Line 96: "... appears to be a threshold concentration...". Corrected
3. Line 484: "... (consistent with our observations) ...". Extra space after the closing parenthesis. Corrected
4. Line 517: "...concentration of mHtt particles that are that are competent...". Corrected
5. Line 545: "...after which cells were reproducibly in exponential growth phase.". Perhaps there is a verb missing before "reproducibly"? Corrected
6. Line 664: "...that the IB has a constant mass density...". Corrected
7. Lines 700-701: "...concentration as the the surface...". Corrected

8. Lines 843-844: part of the title of reference 29 is in uppercase letters. Corrected

Reviewer #3 (Remarks to the Author):

Pei and colleagues set a study to understand in more details how mHtt inclusions are formed in living cell. They chose yeast and mHtt(72Q)-GFP as a model for their study. Using series of microscopic imaging they tracked formation and degradation of mHtt inclusions in living yeast cell. Basing on their data they calculated that new mHtt species seems to be incorporated into the inclusion by random collision and coalescence, rather than active transport as was previously described for the IPOD.

Manuscript is well written with interesting observations. Data generated in this study will be interesting for the community. The experiments are well planned with sufficient explanation in the text. However, there are few points which need to be clarified.

Comments:

1. *Authors have shown that there is a certain threshold for cytoplasmic concentration of mHtt(72Q)-GFP from which IB are formed. Does this correlate with the length of the expressed polyQ region? Authors should compare threshold concentrations of polyQ(72) with longer version of the polyQ expansion.*

Author Response: We thank the reviewers for this suggestion; in order to address this question, we performed long-term time-lapse imaging of cells expressing mHttex1(103Q)-GFP. We find that the growth rate of 103Q IBs is similar to that of 72Q IBs, but that IBs are initiated at lower cytoplasmic concentrations relative to 72Q. There are so few cells with expression levels below the apparent threshold for initiation of an IB in 103Q-expressing cells that we cannot say with confidence that there is a threshold based on these data alone; however, if there is a threshold, it is lower than that of the 72Q construct, as expected for a less stable protein. We've added these findings to the Results and Discussion sections (lines 196-200, Supplementary Note 1, Supplementary Figures 3 and 4).

2. *Figure 4: Authors should include control were proteasome activity or autophagy pathway is blocked to confirm the degradation pathway for mHtt-GFP.*

Author Response: The mechanism of auxin-induced degradation has been well established by many groups, who have shown that AID of a degenon-tagged protein co-expressed with the Tir1 E3 ligase is blocked by a proteasomal inhibitor such as MG132 (Holland et al., 2012; Nishimura et al., 2009; Ramos et al., 2001). MG132 controls have not typically been done in yeast (Kubota et al., 2013; Morawska and Ulrich, 2013; Papagiannakis et al., 2017; Tanaka et al., 2015), possibly because MG132 is not as effective in yeast as mammalian cells: it does not easily pass through the cell wall, and does not completely block degradation by the proteasome, due to the greater importance of the tryptic and caspase-like proteasomal proteases in yeast (Collins et al., 2010).

We have done a set of controls using MG132 to block auxin-mediated degradation in our Tir1, Htt-degron-expressing cells and did see that addition of MG132 appears to block both constitutive degradation seen in Tir1-expressing cells and degradation that was caused by the addition of NAA. The caveat to these experiments is that the SDS treatment necessary to introduce MG132 into the cells dramatically reduced overall cytoplasmic GFP intensity and therefore masked the effect of NAA, resulting in statistically insignificant differences after treatment with NAA, MG132 or both.

The cultures were grown in minimal media made with proline as a nitrogen source and with the addition of low amounts of SDS (0.003%) in order to get MG132 into the cells (Liu et al., 2007; Pannunzio et al., 2004). In previous experiments using conventional minimal medium, addition of 250 μ M NAA resulted in a >5-fold decrease in mean cytoplasmic intensity, from 175 AU to 33 AU. Using proline-based minimal medium with 0.003% SDS, the addition of 250 μ M NAA caused a reduction in mean cytoplasmic intensity from 46 AU to 32 AU.

Within the context of the culture conditions, we did see a restoration of cytoplasmic levels of degron-tagged Htt when the cells were co-incubated with MG132. While the addition of NAA caused a decrease in mean cytoplasmic intensity from 46 AU to 32 AU, incubation with MG132 restored the mean cytoplasmic intensity to 48 AU. The fraction of cells without visible cytoplasmic mHtt increased from 13% to 25% with NAA incubation, but the addition of MG132 to the NAA-treated culture suppressed the effect of NAA, reducing the fraction of cells with no visible cytoplasmic Htt back down to 16%. Unfortunately, due to the low initial cytoplasmic intensities, these differences were not statistically significant (individual pair-wise comparisons done with t-tests, assuming non-Gaussian distribution and unequal variance, n=56-70 cells).

3. *Figure 4: Would inhibition of protein translation in this experiment enhance the clearance of IB?*

Author Response: We feel that specific targeting of Htt for degradation is preferable to a less targeted manipulation that blocks all protein translation. We have applied cycloheximide to Htt-expressing strains and the results were interesting, but difficult to interpret. The fraction of cells with inclusions, particularly abnormal inclusions, rose. In some cells, it appeared that the mHtt was being removed into the vacuole, which is a phenomenon that we normally only see in older, saturated cultures which are under significant metabolic stress (starving). Rising numbers of inclusions and an increase in abnormal inclusion morphology is also typical of cultures after they exit log phase growth due to nutrient deprivation.

The formation of inclusions does require other proteins, including Hsp104, Rnq1 and potentially others. It is modulated by metabolic stress. Comprehensively blocking translation has pleiotropic effects that make the results of the experiment difficult to interpret with confidence. Therefore, we chose to selectively remove mHtt using AID.

4. *Figure S4: The conclusions should be milder as the number of cells analysed is not satisfactory (n=4). Did the authors followed the CLIs over longer time to see if they eventually form a round, IB like structures?*

Author Response: We agree that the relationship between IBs and CLIs is very interesting, and we believe that it deserves more attention. To that end, we have removed the data on CLIs, which we also agree are inadequate for publication at the moment, and are now working on a manuscript that will focus on the connection between IBs and CLIs. We appreciate the suggestion, which has sharpened our attention on this illuminating relationship.

In place of the CLI data, we have added data on auxin-induced degradation of a mHtt(72Q)-GFP construct lacking the proline-rich domain [mHtt(72Q) Δ PRD]. Htt constructs that lack the PRD form numerous abnormal, asymmetric, small inclusions. We added the degron sequence to this construct and found that the mHtt(72Q) Δ PRD-degron-GFP inclusions do not respond to AID. Like the CLI data, these experiments demonstrate that the response to reduced cytoplasmic protein levels is limited to phase-separated IBs. We were easily able to obtain >>4 examples (lines 257-263, Supplementary Note 2, Supplementary Fig 8).

5. *Figure 5: Authors observed that the core particle of IBs stays stable in the cell. What will happen when auxin is washed away? Is the core particle growing back, or is there another IB forming, distinct from the original IB?*

Author Response: This is an important point that we have now addressed; the auxin washout data have been incorporated into a new figure (lines 281-296, Figure 6).

6. *Figure 6: controls should be included showing that the drop in the intensity (for both cells with or without IB) is prevented by the presence of proteasome inhibitor and not by inhibition of autophagy.*

Author Response: As noted above, we did observe that MG132 blocked targeting of degron-tagged Htt to the proteasome.

However, the mechanism of mHtt-GFP degradation in yeast in the absence of degron-tagging is not clearly established. We have observed hundreds of inclusions over a cumulative total of hundreds, possibly thousands, of hours and never observed an instance in which the inclusion disappeared from a cell, so the inclusions are not removed through autophagy.

A visual pulse-chase of mHtt(72Q)-mEos2 suggests that the turnover of cytoplasmic mHtt-mEos2 occurs on a timescale of hours: the reduction in the signal of photoconverted mEos is apparent in under an hour, and the amount of protein in both the cytoplasm and IB continues to decrease over 2- 3 hours after correction for photobleaching and dilution by cell division (Aktar et al., 2019). It is possible that mHtt

is degraded in the IB itself. It is also possible that cytoplasmic mHtt may be removed by the proteasome or by some form of autophagy.

We have attempted to address this question in several ways. First, we have measured the co-localization of mHtt with Atg8, which is involved in production of both autophagosomes and Cvt vesicles (Lynch-Day and Klionsky, 2010; Ohashi and Munro, 2010; Yamasaki and Noda, 2017), and found that mHtt is not associated with Atg8-labeled vesicles. The failure to observe co-localization of mHtt-mCherry with GFP-Atg8 was reported in Aktar et al. (n=694 cells). However, given the limitations of the co-localization assay, we have also inhibited degradation by the vacuole and assessed mHtt-expressing cells for accumulation of vacuolar mHtt-GFP. Vacuolar degradation was reduced by deletion of the gene encoding the vacuolar protease Pep4, which itself is required for maturation of other vacuolar proteinases. GFP-tagged proteins that are targeted to the vacuole can be seen to accumulate in the vacuole in strains lacking Pep4 (Kanki and Klionsky, 2008; Parzych et al., 2018; Schuck et al., 2014).

We imaged *pep4* deletion strains expressing native Httex1(25Q)-GFP, mHttex1(72Q)-GFP or mHttex1(103Q)-GFP. There was no evidence of accumulation of any form of Htt in the vacuole (n = 155, 143 and 63 cells respectively). Just to note, we have seen vacuolar accumulation of Htt-GFP in day-old saturated (nutrient deprived) cultures, so it is possible to observe autophagy of Htt-GFP. But we have not seen evidence of this in healthy, actively growing cells. The *pep4* deletion was confirmed by PCR.

We have also introduced Htt-GFP constructs into *pdr5Δ* cells. Pdr5 is a multidrug transporter, and *pdr5Δ* cells are significantly more susceptible to MG132 treatment. We have seen no effect of the application of 100 μM MG132 to mHtt-GFP cytoplasmic levels or inclusion number. However, there are at least two caveats to this experiment. First, the cytoplasmic levels of mHtt-GFP are normally quite high, so it may be difficult to measure the results of a partial block of the proteasome, the change may be too subtle, especially as we were using an epifluorescence microscope for screening purposes.

Second, the addition of MG132 is known to increase expression of heat shock proteins within 1- 2 hours after addition (Lee and Goldberg, 1998). The increase in chaperones could reduce the amount of unfolded protein that would be targeted to the proteasome, counteracting the effect of the block. Because the addition of cycloheximide has confounding effects (described above in the reply to comment #3), we cannot do a simple pulse-chase.

Lastly, it may be that mHtt is not removed significantly through the proteasome, and that targeting unfolded mHtt-GFP to inclusions is an alternative pathway for its removal. This is an interesting topic that we are actively pursuing using biochemical strategies to enrich for and detect ubiquitinated proteins, which we hope will be more sensitive. We feel that these experiments belong in a manuscript with a focus on Htt turnover.

7. *Figure 7: Does increase in the length of the polyQ expansion increases the growth rate of IBs? Is the growth rate of IBs changed in the condition when cellular active transport is impaired?*

Author Response: We used the timecourse of Httex1(103Q)-GFP-expressing cells to address the first point. The growth of 103Q IBs has the same properties as 72Q IBs, although they form at a lower cytoplasmic threshold and, considering at a particular cytoplasmic intensity, grow faster than 72Q IBs, consistent with a higher proportion of 103Q being unfolded (lines 196-200 and 413-415, Supplementary Notes 1 and 3, Supplementary Figures 4 and 11).

Experiments on strains carrying mutations in molecular motors and expressing mHtt-GFP have been done (Kumar et al., 2016; Wang et al., 2009). Wang and colleagues proposed that Htt aggregates were transported along microtubules to an aggresome located at the microtubule-organizing center (MTOC), based in part on the co-localization of the mHtt aggregate with the MTOC. We have attempted to replicate the co-localization with a variety of MTOC labels and mHtt constructs, but see no evidence of co-localization of inclusions and the MTOC (Aktar et al., 2019). Indeed, as the IB is frequently seen to move around the cytoplasm, that is not really possible.

Kumar and colleagues make a case for actin-based transport of aggregates based on the effect of auxin-mediated degradation of the tropomyosin Tpm2 and the type V myosin yeast Myo2 on inclusions. *Myo2* is an essential gene and mutants exhibit a large range of defects; the Myo2 protein has been shown to interact with 1153 other proteins (<https://www.yeastgenome.org/locus/S000005853>). A 6-hour application of auxin to cells expressing either degron-tagged Myo2 or Tpm2 does indeed result in visible defects in inclusion formation: an increase in the number of large inclusions and/or a change in the morphology of the inclusions. These changes are identical to the changes seen in cells taken from saturated cultures under metabolic duress. The increase in inclusion number is reversible, as are the changes seen in older, saturated cultures when the cells are inoculated into fresh medium (L.E., unpublished data). Six hours corresponds to approximately 3 cell divisions for healthy cells, and *myo2* mutants are defective in cytokinesis (VerPlank and Li, 2005). The cells shown in Kumar et al. are not multi-budded, so they are presumably growing very slowly or not at all.

Thus, there are two arguments against actin-driven transport processes. The first is based on arguments outlined in our previous work: small particles of mHtt-GFP are observed to move completely randomly and the mHtt IB is itself mobile, making it difficult to envision a transport driven growth-mechanism (Aktar et al., 2019). The dynamics of growth, described in the current manuscript, are compatible with a mechanism in which the randomly moving particles collide with the IB and are absorbed.

The second argument against actin-driven processes rests in the paper by Kumar et al. If small aggregates were primarily actively transported to inclusions, the absence of the molecular motor should trigger a buildup of small aggregates throughout the cell, and

inclusions should simply cease to grow. On one hand, the observation that one large inclusion is replaced by multiple large inclusions, or a large abnormal inclusion, does not logically suggest that myosin is required to transport aggregates to inclusions. It suggests a dramatic change in regulation of inclusion formation and structure. On the other hand, the data in Kumar et al. is perfectly compatible with the pleiotropic phenotype of *myo2* mutants and the observation that inclusions in cells under stress both multiply and undergo changes in structure.

Minor comment:

1. In discussion part in lane 441-442 authors suggest that other factors might play a role in IB formation. Could they expand this with some examples? In the revised text we have noted that Hsp104 and Rnq1 are both required for IB formation (line 474). As these proteins are non-essential and their role in inclusion formation discovered by screening candidates, we feel that it is probable that other, as-yet undiscovered proteins also play a role in inclusion initiation and formation.

2. Lane 517: spelling error- 'that are' is used twice. Corrected

3. Figure S2B: fix the graph to have 5h time point. Corrected

References

Aktar, F., Burudpakdee, C., Polanco, M., Pei, S., Swayne, T.C., Lipke, P.N., and Emtage, L. (2019). The huntingtin inclusion is a dynamic phase-separated compartment. *Life Sci. Alliance* 2.

Chen, S., Berthelie, V., Yang, W., and Wetzels, R. (2001). Polyglutamine aggregation behavior in vitro supports a recruitment mechanism of cytotoxicity. *J. Mol. Biol.* 311, 173–182.

Collins, G.A., Gomez, T.A., Deshaies, R.J., and Tansey, W.P. (2010). Combined chemical and genetic approach to inhibit proteolysis by the proteasome. *Yeast Chichester Engl.* 27, 965–974.

Dehay, B., and Bertolotti, A. (2006). Critical role of the proline-rich region in Huntingtin for aggregation and cytotoxicity in yeast. *J. Biol. Chem.* 281, 35608–35615.

Holland, A.J., Fachinetti, D., Han, J.S., and Cleveland, D.W. (2012). Inducible, reversible system for the rapid and complete degradation of proteins in mammalian cells. *Proc. Natl. Acad. Sci.* 109, E3350–E3357.

- Kanki, T., and Klionsky, D.J. (2008). Mitophagy in Yeast Occurs through a Selective Mechanism. *J. Biol. Chem.* *283*, 32386–32393.
- Krishnan, K., and Moens, P.D.J. (2009). Structure and functions of profilins. *Biophys. Rev.* *1*, 71–81.
- Kubota, T., Nishimura, K., Kanemaki, M.T., and Donaldson, A.D. (2013). The Elg1 Replication Factor C-like Complex Functions in PCNA Unloading during DNA Replication. *Mol. Cell* *50*, 273–280.
- Kumar, R., Nawroth, P.P., and Tyedmers, J. (2016). Prion Aggregates Are Recruited to the Insoluble Protein Deposit (IPOD) via Myosin 2-Based Vesicular Transport. *PLOS Genet.* *12*, e1006324.
- Lee, D.H., and Goldberg, A.L. (1998). Proteasome inhibitors cause induction of heat shock proteins and trehalose, which together confer thermotolerance in *Saccharomyces cerevisiae*. *Mol. Cell. Biol.* *18*, 30–38.
- Liu, C., Apodaca, J., Davis, L.E., and Rao, H. (2007). Proteasome inhibition in wild-type yeast *Saccharomyces cerevisiae* cells. *BioTechniques* *42*, 158–162.
- Lynch-Day, M.A., and Klionsky, D.J. (2010). The Cvt pathway as a model for selective autophagy. *FEBS Lett.* *584*, 1359–1366.
- Mahoney, N.M., Janmey, P.A., and Almo, S.C. (1997). Structure of the profilin-poly-L-proline complex involved in morphogenesis and cytoskeletal regulation. *Nat. Struct. Biol.* *4*, 953–960.
- Morawska, M., and Ulrich, H.D. (2013). An expanded tool kit for the auxin-inducible degron system in budding yeast. *Yeast* *30*, 341–351.
- Nishimura, K., Fukagawa, T., Takisawa, H., Kakimoto, T., and Kanemaki, M. (2009). An auxin-based degron system for the rapid depletion of proteins in nonplant cells. *Nat. Methods* *6*, 917–922.
- Ohashi, Y., and Munro, S. (2010). Membrane delivery to the yeast autophagosome from the Golgi-endosomal system. *Mol. Biol. Cell* *21*, 3998–4008.
- Pannunzio, V.G., Burgos, H.I., Alonso, M., Mattoon, J.R., Ramos, E.H., and Stella, C.A. (2004). A Simple Chemical Method for Rendering Wild-Type Yeast Permeable to Brefeldin A That Does Not Require the Presence of an *erg6* Mutation. *J. Biomed. Biotechnol.* *2004*, 150–155.
- Papagiannakis, A., de Jonge, J.J., Zhang, Z., and Heinemann, M. (2017). Quantitative characterization of the auxin-inducible degron: a guide for dynamic protein depletion in single yeast cells. *Sci. Rep.* *7*, 4704.

Parzych, K.R., Ariosa, A., Mari, M., and Klionsky, D.J. (2018). A newly characterized vacuolar serine carboxypeptidase, Atg42/Ybr139w, is required for normal vacuole function and the terminal steps of autophagy in the yeast *Saccharomyces cerevisiae*. *Mol. Biol. Cell* 29, 1089–1099.

Ramos, J.A., Zenser, N., Leyser, O., and Callis, J. (2001). Rapid Degradation of Auxin/Indoleacetic Acid Proteins Requires Conserved Amino Acids of Domain II and Is Proteasome Dependent. *Plant Cell* 13, 2349–2360.

Scherzinger, E., Lurz, R., Turmaine, M., Mangiarini, L., Hollenbach, B., Hasenbank, R., Bates, G.P., Davies, S.W., Lehrach, H., and Wanker, E.E. (1997). Huntingtin-encoded polyglutamine expansions form amyloid-like protein aggregates in vitro and in vivo. *Cell* 90, 549–558.

Schuck, S., Gallagher, C.M., and Walter, P. (2014). ER-phagy mediates selective degradation of endoplasmic reticulum independently of the core autophagy machinery. *J. Cell Sci.* 127, 4078–4088.

Tanaka, S., Miyazawa-Onami, M., Iida, T., and Araki, H. (2015). iAID: an improved auxin-inducible degron system for the construction of a ‘tight’ conditional mutant in the budding yeast *Saccharomyces cerevisiae*. *Yeast* 32, 567–581.

VerPlank, L., and Li, R. (2005). Cell Cycle-regulated Trafficking of Chs2 Controls Actomyosin Ring Stability during Cytokinesis. *Mol. Biol. Cell* 16, 2529–2543.

Walters, R.H., and Murphy, R.M. (2009). Examining polyglutamine peptide length: a connection between collapsed conformations and increased aggregation. *J. Mol. Biol.* 393, 978–992.

Wang, Y., Meriin, A.B., Zaarur, N., Romanova, N.V., Chernoff, Y.O., Costello, C.E., and Sherman, M.Y. (2009). Abnormal proteins can form aggresome in yeast: aggresome-targeting signals and components of the machinery. *FASEB J.* 23, 451–463.

Yamasaki, A., and Noda, N.N. (2017). Structural Biology of the Cvt Pathway. *J. Mol. Biol.* 429, 531–542.

Zhao, Y., Zurawel, A.A., Jenkins, N.P., Duennwald, M.L., Cheng, C., Kettenbach, A.N., and Supattapone, S. (2018). Comparative Analysis of Mutant Huntingtin Binding Partners in Yeast Species. *Sci. Rep.* 8, 9554.

REVIEWERS' COMMENTS:

Reviewer #2 (Remarks to the Author):

In my view, the authors responded well to most comments. This has further strengthened my opinion that the paper is worthy to be published in Communications Biology.

I do have a few remaining points that I hope the authors could consider.

In their response to my major comment #1, the authors had a very elaborate response, partially textually but also experimentally. Regarding the added data on deletion of the Δ PRD construct with a degron in between Htt and GFP, they found that the protein did form inclusions but are not phase separated. Whilst this answers my question as such, it however also provokes the question as to how general the mechanism described here on inclusion formation (growth of the inclusion occurs through collision and coalescence of small, aggregative particles into a suspension of particles that is phase separated from the surrounding cytoplasm) is and applied to other polyQ diseases (or even more so other disease showing inclusion formation). I think they authors should point this out in their discussion.

Furthermore, I am somewhat disappointed of not responding tot my request to repeat at least a few key experiments with strains lacking Hsp104. In my view, this would have made the paper more interesting. Without this, the lines on this matter in the discussion should then better be deleted in my view.

Reviewer #3 (Remarks to the Author):

The additional experiments and extensive explanation provided by the authors are fully satisfactory and therefore, I recommend this manuscript for publication.

We'd like to thank the reviewers for their constructive feedback and positive comments. Reviewer #2 has drawn our attention back to larger questions regarding inclusion formation, as well as expressing frustration at the lack of data on inclusion growth in the absence of Hsp104. With regard to the first point, we hope that this study contributes to characterizing the variety of inclusion types in greater detail, and also to increasing our understanding of the mechanism underlying the formation and growth of phase-separated inclusions in particular. We believe that compelling questions of inclusion formation in vivo must be addressed through careful study of the properties of particular inclusion classes before generalizations can be made.

With regard to the second point, this is a question that cannot be experimentally addressed as cells lacking Hsp104 do not form inclusions or visible aggregates (expanded on below).

Reviewer#2: In their response to my major comment #1, the authors had a very elaborate response, partially textually but also experimentally. Regarding the added data on deletion of the ΔPRD construct with a degron in between Htt and GFP, they found that the protein did form inclusions but are not phase separated. Whilst this answers my question as such, it however also provokes the question as to how general the mechanism described here on inclusion formation (growth of the inclusion occurs through collision and coalescence of small, aggregative particles into a suspension of particles that is phase separated from the surrounding cytoplasm) is and applied to other polyQ diseases (or even more so other disease showing inclusion formation). I think they authors should point this out in their discussion.

Author's response: Thank you for raising this point, we wholeheartedly agree that generalizations about the mechanism of inclusion growth are currently not possible. With regard to ΔPRD inclusions, they do appear to be less soluble, possibly solid, and solid inclusions are quite likely to grow through collision and accretion. However, we currently have no evidence for the mechanism of their growth. The reason that mHtt constructs with and without the PRD form very different inclusion types is also an interesting and currently unanswered question. Generally, we agree that it is critical to more fully characterize the variety of inclusion structure and pathways of formation and have expanded upon this point in the Discussion (lines 584-8).

Reviewer#2: Furthermore, I am somewhat disappointed of not responding tot my request to repeat at least a few key experiments with strains lacking Hsp104. In my view, this would have made the paper more interesting. Without this, the lines on this matter in the discussion should then better be deleted in my view.

Author's response: We thank Reviewer #2 for bringing this to our attention. To clarify, we cannot probe mutant Httex1-GFP inclusions in strains lacking Hsp104 because *hsp104* deletion strains do not form visible inclusions or aggregates of any kind; these observations have been published with regard to the full Htt exon 1 construct (Krobitsch and Lindquist, 2000, DOI: 10.1073/pnas.97.4.1589, Aktar et al., 2019, DOI: 10.26508/lsa.201900489). We have added a sentence to the Introduction in order to make this point more explicit (lines 83-5) and noted this in the Discussion (lines 488, 576-8 and 581-2).